# Retrieval of aerosol composition from spectral aerosol optical depth and optical properties using a machine learning approach

Denghui Ji[1], Xiaoyu Sun[1], Christoph Ritter[2], and Justus Notholt[1]

[1]Institute of Environmental Physics, University of Bremen, Otto-Hahn-Allee 1, 28359 Bremen, Germany
[2]Alfred Wegener Institute, Helmholtz Centre for Polar and Marine Research, Telegrafenberg A43, 14473 Potsdam, Germany

**Correspondence:** Xiaoyu Sun (xiaoyu_sun@iup.physik.uni-bremen.de)

**Abstract.** Accurate aerosol composition retrievals support radiative forcing assessment, source attribution, air quality analysis, and improved modeling of aerosol–cloud–radiation interactions. Aerosol retrievals based solely on visible-wavelength aerosol optical depth (AOD) observations provide limited spectral sensitivity, which may be insufficient to reliably distinguish among aerosol types with similar optical properties. In this study, we present a new retrieval framework that combines multi-wavelength AOD observations from both the visible and shortwave infrared spectrum, enhancing aerosol type discrimination. A neural network forward model trained on simulations from the Model for Optical Properties of Aerosols and Clouds (MOPSMAP), which relates aerosol optical properties to spectral AOD, is embedded in an optimal estimation method (OEM) to retrieve aerosol composition. This machine learning-based forward model achieves computational efficiency without making compromises in accuracy. The neural network forward model achieves a mean $R^2$ of 0.99 with root-mean-square error below 0.01. The retrieval resolves up to four independent aerosol components, with degrees of freedom for signal about 3.75. We apply this hybrid method to ground-based observations, including data from the Aerosol Robotic Network (AERONET) and Fourier Transform Infrared spectrometer (FTIR) measurements. The retrieved aerosol compositions are consistent with physical expectations and validated through backward trajectory analysis.

## 1 Introduction

Aerosols play an important role in the climate system by influencing the Earth's radiation budget (Kuniyal and Guleria, 2019; Haywood, 2021), cloud microphysics (Mauritsen et al., 2011; Gong et al., 2023), and air quality (Garrett and Zhao, 2006). Depending on their properties, aerosols can either cool the Earth's surface by reflecting incoming solar radiation (Charlson and Wigley, 1994; Chang et al., 2022), or warm the atmosphere by absorbing sunlight (Weinbruch et al., 2012; Bond et al., 2013; Breider et al., 2014; Groot Zwaaftink et al., 2016; Kodros, 2018). While the net global effect of aerosols is cooling, their climatic impact varies significantly with aerosol type, spatial distribution, and environmental conditions (Kaufman et al., 2002; Satheesh and Moorthy, 2005). For example, strongly scattering aerosols such as sulfate and sea salt typically have a

cooling effect in lower-latitude regions. However, in the Arctic, sea salt aerosols can undergo hygroscopic growth under Arctic humidity, which enhances their infrared radiative properties (Ji et al., 2025) and potentially contributes to the longwave cloud radiative effects (Gong et al., 2023). These complexities highlight the importance of accurately observing aerosol microphysical and optical properties in different environmental conditions to better quantify the impact of aerosols on climate and improve the performance of climate models.

Aerosol optical depth (AOD) is a fundamental parameter used to describe the column-integrated extinction of solar radiation due to aerosols. AOD retrievals can be obtained through both active and passive remote sensing techniques. Active remote sensing methods, such as lidar, provide vertically resolved aerosol properties and have been widely used on both ground-based and satellite platforms (Jin et al., 2020; Floutsi et al., 2023). For instance, the Cloud-Aerosol Lidar and Infrared Pathfinder Satellite Observation (CALIPSO) mission provides detailed aerosol vertical distributions, offering crucial insights into aerosol transport and layering (Winker et al., 2007; Liu et al., 2009). Passive remote sensing, on the other hand, relies on the measurement of scattered and absorbed radiation and includes both satellite-based and ground-based instruments. Satellite sensors such as the Moderate Resolution Imaging Spectroradiometer (MODIS) retrieve AOD on global scales using multi-spectral radiance measurements (Levy et al., 2007), while ground-based networks like the AErosol RObotic NETwork (AERONET) provide high-accuracy AOD measurements at multiple wavelengths through sun photometry (Holben et al., 1998; Giles et al., 2019b).

Despite the abundance of AOD observations, retrieving aerosol composition from remote sensing remains challenging. Recent studies have introduced new methods for retrieving aerosol composition (Li et al., 2019; Ji et al., 2023). In particular, Fourier Transform Infrared spectrometer (FTIR) has been successfully employed to extract aerosol component information from infrared emission spectra (Ji et al., 2023). This method provides valuable insights into aerosol microphysical and chemical properties. The incorporation of shortwave infrared spectral information into aerosol retrieval algorithms offers a promising method for improving the accuracy of aerosol composition estimation. Barreto et al. (2020) and Alvárez et al. (2023) have established a detailed observation framework that combines AERONET and FTIR measurements to obtain aerosol AOD spectra spanning both visible and shortwave infrared wavelengths. Despite the availability of such comprehensive spectral observations, no existing retrieval algorithm has been developed to infer aerosol composition based on joint visible–shortwave-infrared AOD data. This study aims to fill that gap.

In aerosol remote sensing, radiative transfer models and aerosol optical property calculators are fundamental to developing a full-physics retrieval algorithm. For example, MODIS aerosol retrievals use look-up tables based on radiative transfer simulations (Levy et al., 2007), while AERONET applies a detailed multi-wavelength approach to observe aerosol size distribution and refractive index (Giles et al., 2019b). However, the complex dependence of aerosol optical properties on size distribution, composition, relative humidity, and multiple scattering introduces strong nonlinearity into the aerosol retrievals, making traditional retrievals computationally intensive and challenging to optimize. To address these challenges, machine learning (ML) methods have emerged as promising alternatives, offering the potential to approximate the nonlinear mappings between aerosol properties and observations more efficiently while retaining the underlying physical constraints learned from full-physics simulations.

In recent years, machine learning techniques have been widely explored to enhance remote sensing retrievals, offering substantial improvements in efficiency and data assimilation (Cobb et al., 2019; Himes et al., 2020; Doicu et al., 2021; Tian and Shi, 2022; Li et al., 2023). However, ML models are often criticized for their lack of physical interpretability, functioning as

"black-box" algorithms without explicit ties to underlying atmospheric physics. Despite these limitations, some studies have demonstrated the potential of ML to replace specific components of physical models (Himes et al., 2020). For example, a hybrid radiative transfer and transfer learning framework is proposed to retrieve aerosol optical depth and fine-mode fraction from multi-spectral geostationary satellite data (Tang et al., 2025). Additionally, neural network–based retrieval approaches using TROPOMI $O_2$ A-band spectra have been developed for aerosol parameter inference (Rao et al., 2022), and radiative

transfer emulators have been integrated into TROPOMI aerosol layer height algorithms (Nanda et al., 2019). The FastMAPOL algorithm employs neural network-based forward models within a multi-angle polarimetric retrieval framework, achieving speed-ups of about 1000× with minimal accuracy loss (Gao et al., 2021a). It also includes adaptive view-angle filtering to mitigate retrieval errors from problematic geometries in satellite and airborne data (Gao et al., 2021b). Similarly, the PACE-MAPP algorithm couples atmosphere–ocean vector radiative transfer emulators to jointly retrieve aerosol and ocean optical properties

from polarimetric measurements (Stamnes et al., 2023). In addition, algorithms such as the Generalized Retrieval of Aerosol and Surface Properties (GRASP) algorithm (Dubovik et al., 2011a) and the Remote Sensing of Trace Gases and Aerosol Products (RemoTAP) algorithm (Hasekamp et al., 2021) have integrated radiative transfer emulation strategies and have been widely applied to global aerosol data from POLDER and PACE (Hasekamp et al., 2024).

These examples highlight the growing role of machine learning emulators in operational and research retrieval systems.

While most prior work focuses on improving radiative transfer speed or expanding polarimetric capabilities, our approach extends the emulator concept to direct aerosol composition retrieval from visible and shortwave infrared AOD spectra, emphasizing feasibility in ground-based and satellite multi-band systems.

The ultimate goal of this study is to develop an algorithm to retrieve global aerosol composition from AOD observations at visible and shortwave infrared wavelengths. Traditionally, this retrieval relies on constructing a relationship between AOD and

80 aerosol composition using full-physics models, such as the Model of Optical Properties of Aerosols and Clouds (MOPSMAP, (Gasteiger and Wiegner, 2018)). In this study, MOPSMAP is used to generate a training dataset, and a ML model is trained to capture the mapping between AOD and aerosol composition. The trained ML model then serves as a forward model, replacing the traditional physical model in the inversion process. This approach can approximate the full-physics forward model with a faster, data-driven algorithm that can be applied globally.

In Section 2, we describe aerosol datasets used in this study, including both ground-based measurements and satellite observations. We present the construction steps of the ML database, the training process, and how it is integrated into the retrieval algorithm in Sect. 3. Section 4 presents the results, followed by a discussion on the implications and limitations of the proposed approach.

## 2 Data

### 2.1 Multi-band AOD Measurements from AERONET and FTIR

In this study, ground-based measurements are conducted in Ny-Ålesund (11.5° E, 78.9° N), including a sun photometer (AERONET) and a Fourier Transform Infrared spectrometer. The FTIR system (Notholt et al., 1995) is a Bruker 120HR instrument operated as part of the Network for the Detection of Atmospheric Composition Change (NDACC). FTIR leads high resolution, $0.0035\,\mathrm{cm}^{-1}$, spectra in infrared. Barreto et al. (2020) and Alvárez et al. (2023) provide a detailed methodology, the Langley calibration method, for measuring AOD using the shortwave infrared spectrum from FTIR. Following their approach, this study derives aerosol AOD observations in the spectral range, including 1020.90, 1238.25, 1558.25, 2133.40, 2192.00, and 2314.20 nm.

Standard AERONET sun photometers retrieve AOD at 340, 380, 440, 500, 675, 870, 1020, and 1640 nm, covering the ultraviolet (UV) to shortwave infrared (SWIR) range (Floutsi et al., 2023). In addition to these direct sun measurements, AERONET provides inversion products that include single scattering albedo (SSA) at 440 nm, asymmetry factor at 440 nm (AF), and effective radius (Reff), retrieved using sky radiance observations (Dubovik and King, 2000; Giles et al., 2019a). These parameters are useful for aerosol type discrimination and are used in this study.

In Ny-Ålesund, the selected wavelengths start from 440 nm, as shorter wavelengths (340 and 380 nm) are not available. In summary, based on the combined AERONET and FTIR observations, the aerosol optical depth (AOD) retrievals in this study are performed at the following wavelengths: 440, 550, 675, 870, 1020, 1558, and 2192 nm.

### 2.2 MERRA-2 aerosol reanalysis data

The Modern-Era Retrospective Analysis for Research and Applications version 2 (MERRA-2) is the latest global atmospheric reanalysis by NASA's Global Modeling and Assimilation Office (GMAO) using the GEOS atmospheric model (version 5.12.4). MERRA-2 provides a physically consistent, long-term record of meteorological and aerosol variables from 1980 to the present (Gelaro et al., 2017).

In this study, we use the MERRA-2 aerosol product M2T1NXAER (also referred to as tavg1_2d_aer_Nx), which is an hourly, time-averaged, two-dimensional dataset. This product includes assimilated aerosol diagnostics such as column-integrated AOD at 550 nm. The aerosol species in MERRA-2 include black carbon, dust, sea salt, sulfate, and organic carbon. The aprior information used in this study is the relative number concentration fractions of individual aerosol components. The derivation of these fractions from MERRA-2 aerosol optical depth data is detailed in Sect.3.7.

## 3  Method

### 3.1  Overview of the Methodological Framework

This study presents a hybrid framework that combines physics-based aerosol optical property calculator , machine learning (ML), and optimal estimation method (OEM) to retrieve aerosol composition from multi-wavelength AOD observations. The key idea is to reorganize the physical input-output structure of forward model (MOPSMAP) to align with what is actually observable.

Instead of directly retrieving aerosol properties from a full-physics model, we first generate a synthetic database using MOPSMAP, in which each sample includes both microphysical inputs and resulting aerosol optical outputs. From this database, we construct a new forward model using machine learning. Specifically, the ML model takes as input the aerosol component fractions (i.e., number concentrations of sea salt, sulfate, black carbon, dust, and insoluble aerosols), together with auxiliary parameters such as single scattering albedo (SSA), asymmetry factor (AF), and effective radius (Reff). These parameters are not routinely available from all remote sensing observations: they are only sometimes retrievable from ground-based measurements under specific viewing and scene conditions, and are rarely robustly retrieved from satellite observations, particularly in low-AOD scenes or in cases where the aerosol field is spatially heterogeneous. The output of the model is the spectral aerosol optical depth (AOD), originally computed by MOPSMAP. Once trained, this ML-emulated forward model is embedded within an optimal estimation framework. Observed spectral AOD (e.g., from ground-based measurements or satellite retrievls) is used as the retrieval input, and aerosol component fractions are estimated iteratively by minimizing the mismatch between observed and simulated AOD, under constraints from a prior information and measurement uncertainty.

Specifically, as shown in Fig. 1, MOPSMAP takes as input a variety of aerosol microphysical parameters, including component fractions, particle size distribution, complex refractive index, particle shape, and ambient relative humidity. It then computes corresponding optical properties, such as SSA, AF, and Reff, which are usually observable using instruments such as sun photometers. We therefore reorganize the MOPSMAP simulation inputs and outputs according to actual observational conditions. In summary, part of the original MOPSMAP outputs (SSA, AF, and Reff) are repurposed as inputs to a machine learning model. This is not achievable by the traditional forward simulation itself, but can be enabled by data-driven learning.

To implement the proposed aerosol composition retrieval framework, we follow a structured approach consisting of three main steps. These are outlined as follows and will be described in detail in the following sections:

1. **Synthetic Dataset Generation:** A large AOD dataset is generated using MOPSMAP by varying aerosol component fractions, as well as four physically-constrained parameters: single scattering albedo (SSA), asymmetry factor (AF), effective radius (Reff), and relative humidity (RH).

2. **Machine Learning Forward Model:** A neural network is trained to emulate a new forward model, mapping input parameters (the aerosol component fractions, SSA, AF, and Reff) to multi-wavelength AOD spectra.

3. **Retrieval via Optimal Estimation:** The ML-based forward model is integrated into an optimal estimation framework to retrieve aerosol composition from observed AOD.

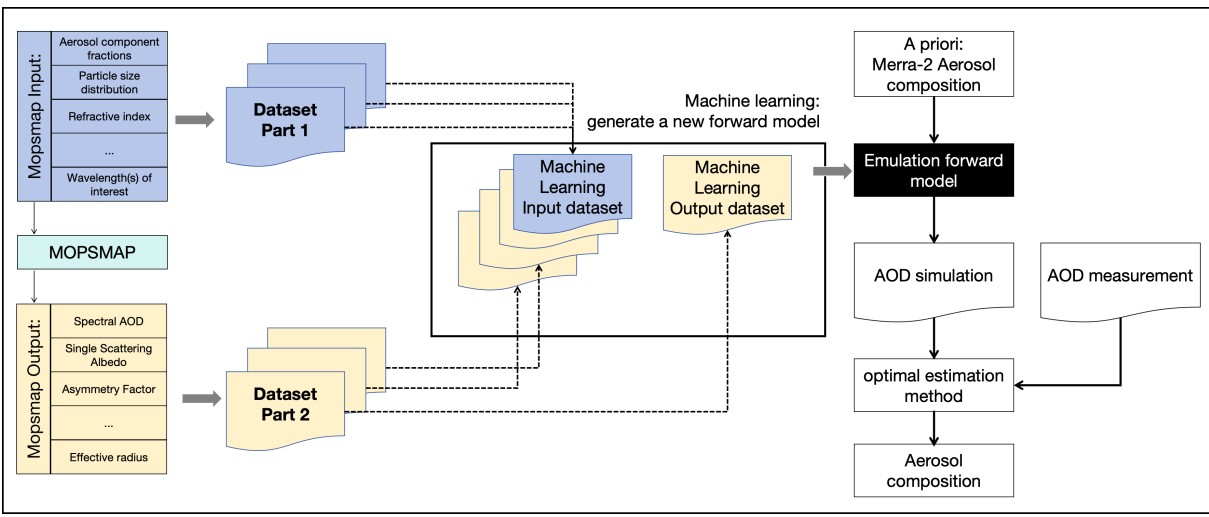

**Figure 1.** Schematic overview of the model development and retrieval workflow. Left panel: MOPSMAP is used to generate a synthetic database by simulating optical properties from randomly sampled aerosol parameters, including component fractions, size distribution, and refractive index. Based on the subset of optical parameters that are typically available in real observations (e.g. SSA, AF, and Reff), we reorganize the simulated database to define the machine learning inputs. The corresponding aerosol component fractions, SSA, AF, and Reff are selected as the machine learning inputs. The spectral AOD is the machine learning output. This effectively inverts the original MOPSMAP input-output structure to train an efficient, observation-driven emulator of the forward model. Right panel: The trained ML-based forward model is then used within an optimal estimation framework. Prior aerosol composition is taken from MERRA-2 (Gelaro et al., 2017), and AOD is simulated using the learned forward model. By minimizing the mismatch with observed AOD measurements, the aerosol composition is retrieved. This setup allows flexible integration of real-world measurements and efficient inversion without full-physics simulation during retrieval.

## 3.2 Aerosol Optical Database Simulation with MOPSMAP

The Model for Optical Properties of Aerosols and Clouds (MOPSMAP, (Gasteiger and Wiegner, 2018)) is a numerical tool designed to compute aerosol and cloud optical properties based on Lorenz-Mie theory and the T-matrix method. It supports a wide range of aerosol compositions, including sulfates, sea salt, black carbon, mineral dust, and organic aerosols, with flexible size distributions (e.g., log-normal, gamma) and shape assumptions (spherical and non-spherical particles). Covering a broad spectral range from ultraviolet (UV) to thermal infrared (IR), MOPSMAP provides key optical parameters such as extinction,

scattering, and absorption coefficients, single scattering albedo (SSA), asymmetry parameter, and phase functions.

To construct a comprehensive dataset for training a machine learning model, we use the MOPSMAP for aerosol optical property simulations. We consider five primary aerosol composition, similar to MERRA-2 (The Modern-Era Retrospective analysis for Research and Applications, Version 2) reanalysis data (Gelaro et al., 2017): sea salt, sulfate, black carbon, dust, and insoluble aerosols. The size distributions of dry aerosols follow a log-normal distribution, with sea salt and dust ranging

from 0.01 to 0.5 μm, while sulfate, black carbon, and insoluble aerosols range from 0.01 to 0.1 μm. Varying proportions of

**Table 1.** Overview of input parameters and simulated outputs used in the MOPSMAP-based aerosol optical property database.

| Category | Parameter | Description |
|---|---|---|
| **Input Parameters** | Aerosol Composition | Fractions of sea salt, sulfate, black carbon, dust, and insoluble; sum to 100% |
| | Size Distribution | Log-normal; sea salt and dust: 0.01–0.5 μm; others: 0.01–0.1 μm (dry mode) |
| | Relative Humidity (RH) | Varied from 55% to 95%, affecting wet particle size |
| | Particle Shape | spherical |
| **Output Variables** | AOD Spectrum | Extinction AOD at 440, 550, 675, 870, 1020, 1558, and 2192 nm |
| | Single Scattering Albedo (SSA) | at 440 nm for training |
| | Asymmetry Factor (AF) | at 440 nm for training |
| | Effective Radius (Reff) | Wet effective radius derived from size and RH |

these five aerosol classes are randomly assigned, with their total constrained to sum to 100%. This process is repeated 10,000 times, with each time a randomly varying proportion of five aerosols, to produce a dataset that covered a wide range of aerosol mixtures commonly found in the atmosphere. Hygroscopic growth significantly alters aerosol optical properties, particularly in the shortwave infrared spectrum. We incorporate this effect by varying the relative humidity (RH) between 55% and 95% in the simulations. The optical properties of the aerosol mixtures are then computed under these conditions.

As given in Tab.1, using MOPSMAP, we simulate 10,000 cases, obtaining a set of aerosol optical properties across multiple wavelengths, including: Extinction coefficient at visible and shortwave infrared wavelengths (440, 550, 675, 870, 1020, 1558, 2192 nm), Single scattering albedo (SSA), Asymmetry factor (AF) and Effective radius (Reff). To focus on the spectral shape of AOD rather than absolute magnitude, we scale all wavelengths relative to the 440 nm value. This results in a relative AOD spectrum, with AOD (440 nm) set to 1.0 and all other wavelengths scaled accordingly. The retrieved outputs represent number concentration fractions of aerosol components. Then, in post-processing, these fractions can be combined with observed or modeled AOD magnitudes to calculate component-specific AODs. However, due to differences in size-dependent extinction efficiency among aerosol types, the AOD contribution of each component is not strictly proportional to number fraction. Therefore, while the absolute AOD values of individual components may have some uncertainty, the spatial distribution and relative dominance of each aerosol type remain meaningful, particularly in high-AOD cases. The script used to generate this database is available in Section data and code.

### 3.3 Neural Network Model Training

Directly using MOPSMAP in the retrieval is challenging, likely due to the high dimensionality of its input parameters and the strong nonlinearity in the model. Therefore, to replace MOPSMAP as a forward model, we develop a machine learning framework that learns the relationship between aerosol composition and its optical properties. The trained model allows for

rapid calculations in retrieval applications. All detailed procedures and comments are available in the accompanying code repository (see "code data availability"). Below, we provide a brief summary of the training workflow.

The input features consist of 5 aerosol component fractions (sea salt, sulfate, black carbon, dust, and insoluble), along with 4 physically-constrained parameters: SSA, AF, Reff, and RH. The output labels are the corresponding AODs at eight wavelengths: 440, 500, 550, 675, 870, 1020, 1558, and 2192 nm. We adopt a fully connected feed-forward neural network with two hidden layers, which is commonly used in aerosol retrieval studies (Faure et al., 2001; Nanda et al., 2019; Chen et al., 2022). The architecture is defined as:

$$\text{Input } (9) \rightarrow \text{FC}(32) \rightarrow \text{ReLU} \rightarrow \text{FC}(32) \rightarrow \text{ReLU} \rightarrow \text{FC}(9) \rightarrow \text{Output (AOD)} \tag{1}$$

Here, $\text{FC}(n)$ denotes a fully connected layer with $n$ neurons. Each hidden layer is followed by a rectified linear unit (ReLU) activation function (Nair and Hinton, 2010), defined as:

$$\text{ReLU}(z) = \max(0, z) \tag{2}$$

ReLU introduces non-linearity into the model, is computationally efficient, and helps mitigate the vanishing gradient problem, enabling effective training of deep networks.

To ensure generalization and avoid overfitting, the dataset is split as:

$$\text{Train} : \text{Validation} : \text{Test} = 70\% : 15\% : 15\% \tag{3}$$

The model is trained using the Adam optimizer (Kingma and Ba, 2017) with a learning rate of $10^{-3}$ and batch size of 64. Performance is evaluated using the mean squared error (MSE) as the loss function:

$$\text{MSE} = \frac{1}{N} \sum_{i=1}^{N} (Y_{\text{true}}^{(i)} - Y_{\text{pred}}^{(i)})^2 \tag{4}$$

Model accuracy is assessed with the root mean square error (RMSE) and coefficient of determination ($R^2$):

$$R^2 = 1 - \frac{\sum (Y_{\text{true}} - Y_{\text{pred}})^2}{\sum (Y_{\text{true}} - \bar{Y})^2} \tag{5}$$

$$\text{RMSE} = \sqrt{\frac{1}{N} \sum (Y_{\text{true}} - Y_{\text{pred}})^2} \tag{6}$$

By using this machine learning-based forward model, we achieve a computationally efficient alternative to MOPSMAP, making it feasible for large-scale aerosol composition retrievals from both ground-based and satellite measurements. This approach not only reduces computational cost but also preserves the essential physical relationships governing aerosol optical properties, enabling large-scale and physically consistent aerosol composition retrievals in the subsequent optimal estimation framework.

## 3.4 Aerosol Composition Retrieval Using Optimal Estimation

To retrieve aerosol composition from multi-wavelength AOD measurements, we apply the optimal estimation method (OEM) (Rodgers, 2000). The key idea is to iteratively adjust the aerosol composition vector until the simulated AOD spectrum matches the observed one, under physical constraints provided by prior knowledge and measurement uncertainty.

In traditional full-physics approaches, such as using MOPSMAP directly, the state vector $\mathbf{x}$ may include high-dimensional microphysical properties like aerosol size distribution, number concentration, and refractive index for each aerosol component:

$$\mathbf{x} = [n_1(r), N_1, \ldots, n_5(r), N_5]^T \tag{7}$$

where $n_i$ and $N_i$ are the aerosol size distribution and number concentration of the five aerosol types: sea salt, sulfate, soot, dust, and insoluble aerosols. $n_i(r)$ denotes the log-normal size distribution for component $i$, defined as:

$$n_i(r) = \frac{N_i}{\sqrt{2\pi}, \ln\sigma, r} \exp\left[-\frac{1}{2}\left(\frac{\ln r - \ln r_{\text{mod}}}{\ln\sigma}\right)^2\right] \tag{8}$$

$r_{mod}$ and $\sigma$ are internal parameters of the log-normal distribution of the aerosol. The MOPSMAP captures physical details in aerosol optical properties but could result in ill-posed inverse problems. In practice, we have also implemented a full-physics optimal estimation algorithm based on MOPSMAP directly; however, due to poor retrieval convergence and unstable performance, this approach is not further considered in the current study.

However, if we have reconstructed MOPSMAP using machine learning, the trained model can greatly simplify the input parameters and can guarantee the accuracy of the output aerosol AOD simulation, then the state vector can be reduced from nearly ten dimensions to five dimensions:

$$\mathbf{x} = [N_1, N_2, N_3, N_4, N_5]^T \tag{9}$$

ML makes a clever connection between all input parameters to have only 5 parameters, which is easier to converge. Thus, the nonlinearity of the inversion process can be reduced, and the accuracy and speed of the inversion can be improved. This model approximates the forward mapping from aerosol composition to AOD as:

$$\mathbf{y} = f(\mathbf{x}; \boldsymbol{\theta}) \tag{10}$$

where $\mathbf{x} \in \mathbb{R}^5$ represents the aerosol component fractions (sea salt, sulfate, BC, dust, and insoluble), and $\boldsymbol{\theta} = \{\text{SSA}, \text{AF}, \text{Reff}, \text{RH}\}$ are fixed auxiliary parameters that encode environmental and optical conditions. The model $f$ is learned from a large MOPSMAP-generated dataset and replaces the computationally intensive bulk aerosol optical property calculator step.

The OEM retrieves $\mathbf{x}$ by minimizing a cost function that balances fidelity to the observed AOD spectrum, $\mathbf{y}_{\text{obs}}$, and deviation from a prior estimate $\mathbf{x}_a$:

$$J(\mathbf{x}) = (\mathbf{y}_{\text{obs}} - f(\mathbf{x}; \boldsymbol{\theta}))^T \mathbf{S}y^{-1}(\mathbf{y}_{\text{obs}} - f(\mathbf{x}; \boldsymbol{\theta})) + (\mathbf{x} - \mathbf{x}_a)^T \mathbf{S}_a^{-1}(\mathbf{x} - \mathbf{x}_a) \tag{11}$$

Here, $\mathbf{S}_a$ is the a prior covariance matrix, and $\mathbf{S}_y$ is the observation error covariance derived from AOD measurement uncertainty. The a prior vector $\mathbf{x}_a$ is derived from the MERRA-2 monthly mean aerosol component fractions at the same time and location as the FTIR observations. The a priori covariance matrix $\mathbf{S}_a$ is specified as a diagonal matrix. For the synthetic (virtual-spectrum) experiments, a uniform variance of $0.01$ (i.e., a standard deviation of $0.1$) is prescribed for each aerosol component. For retrievals using actual observations, the diagonal elements of $\mathbf{S}_a$ are derived empirically from the spatial variability of MERRA-2 aerosol component fractions within a $\pm 1°$ latitude-longitude box centered on the observation location, and are used as aprior uncertainties. For the measurement error covariance matrix $\mathbf{S}_y$, we distinguish between visible and shortwave infrared (SWIR) wavelengths. For visible bands, we adopt wavelength-dependent AOD uncertainties following AERONET direct-Sun uncertainty estimates, with a standard deviation of $0.02$ at 440 nm and shorter wavelengths, and $0.01$ at longer visible wavelengths. For the infrared bands (SWIR), we adopt $0.02$ as the standard deviation, based on reported uncertainties from Barreto et al. (2020) and Alvárez et al. (2023), who applied Langley calibration for FTIR-based AOD measurements in the SWIR region. All uncertainties are assumed to be spectrally uncorrelated.

The state vector is updated iteratively using the Gauss-Newton method. The Jacobian matrix $\mathbf{K}$, representing the sensitivity of AOD to changes in aerosol components, is numerically computed via finite differences:

$$\mathbf{K} = \frac{\partial f(\mathbf{x}; \boldsymbol{\theta})}{\partial \mathbf{x}} \tag{12}$$

The gain matrix $\mathbf{G}$ and update equation are:

$$\mathbf{G} = (\mathbf{K}^T \mathbf{S}_y^{-1} \mathbf{K} + \mathbf{S}_a^{-1})^{-1} \mathbf{K}^T \mathbf{S} y^{-1} \tag{13}$$

$$\mathbf{x}_{n+1} = \mathbf{x}_n + \mathbf{G}(\mathbf{y}_{\text{obs}} - f(\mathbf{x}_n; \boldsymbol{\theta})) \tag{14}$$

where $n$ is the iteration index.

This hybrid retrieval framework reduces computational cost and avoids non-convergence issues common in full-physics OEMs, while maintaining physical realism through the machine-learned forward operator and inclusion of environmental parameters as constraints.

### 3.5 Uncertainty Analysis

As we mentioned before, a comprehensive virtual database is constructed, covering a wide range of aerosol compositions. To quantitatively assess retrieval uncertainty, we avoid relying solely on limited ground-based observations, which may not be representative. Instead, a total of 1500 cases are randomly selected (15% of the full 10,000-sample dataset, they do not represent a designated "test" set.) and the corresponding AOD spectrum are used as the synthetic AOD observation. For each selected sample, we perturb the aerosol component by 10% (acted as a priori), followed by normalization. Subsequently, these 1,500 cases are processed through the full retrieval procedure. This experimental configuration facilitates a systematic and controlled evaluation of the retrieval algorithm under diverse aerosol scenarios, thereby supporting a robust assessment of error characteristics and retrieval performance.

The posterior error covariance matrix $\mathbf{S}_{\text{post}}$ for each sample is given by:

$$\mathbf{S}_{\text{post}} = \left( \mathbf{K}^{\text{T}} (\mathbf{S}_y + \mathbf{S}_f)^{-1} \mathbf{K} + \mathbf{S}_a^{-1} \right)^{-1} \tag{15}$$

where $\mathbf{K}$ is the Jacobian matrix, estimated numerically by finite perturbations; $\mathbf{S}_y$ is the measurement error covariance matrix; $\mathbf{S}_f$ is the forward model error covariance matrix, estimated as the squared residuals between simulated and predicted AOD spectra, $\mathbf{S}_a$ is the prior error covariance matrix. The total retrieval uncertainty for each aerosol type is decomposed into three components: observation error contribution, a prior error contribution, and forward model error contribution. Finally, the result averaged over the 1500 cases allows us to quantify the dominant sources of uncertainty for each aerosol component.

To further assess the information content of the retrieval system, we calculate the Averaging Kernel (AVK) matrix $\mathbf{A}$, defined as:

$$\mathbf{A} = \frac{\partial \mathbf{x}_{\text{retrieved}}}{\partial \mathbf{x}_{\text{true}}} = \mathbf{GK} \tag{16}$$

The diagonal elements of the AVK matrix indicate the degree to which each aerosol component is constrained by the observations. The trace of the AVK matrix gives the Degrees of Freedom for Signal (DoF):

$$\text{DoF} = \text{trace}(\mathbf{A}) \tag{17}$$

indicating how much independent information is effectively retrieved from the measurement. This highlights the potential and limitation of our retrieval algorithm in distinguishing aerosol types under realistic error assumptions.

### 3.6 Sensitivity analysis to auxiliary parameters

To assess the impact of uncertainties in auxiliary optical parameters on the aerosol composition retrieval, we conducted a sensitivity analysis using the trained ML forward model. Aerosol composition fractions are retrieved from multi-wavelength AOD observations, while the parameters, SSA, AF, Reff, and RH, are treated as auxiliary inputs. In the retrieval, these auxiliary parameters are assumed to be fixed but subject to certain uncertainty. In the following, we quantify how uncertainties in these auxiliary parameters propagate into the simulated AOD spectrum and the retrieved results of aerosol composition.

We adopt representative uncertainty ranges of $\pm 0.03$ for SSA, $\pm 0.05$ for AF, $\pm 20\%$ for Reff, and $\pm 5\%$ for RH. These values are consistent with typical AERONET uncertainty estimates for SSA and with commonly reported uncertainties for aerosol optical properties (e.g., Dubovik et al. (2002)). The impact of auxiliary-parameter uncertainty is quantified through an ensemble-based retrieval experiment designed to propagate realistic parameter errors. The four auxiliary parameters (SSA, AF, Reff, and RH) are perturbed simultaneously according to Gaussian distributions defined by their assumed uncertainty ranges, and the optimal estimation retrieval is repeated for each realization. A total of 300 realizations are performed to characterize the resulting variability in the retrieved aerosol composition. The resulting ensemble of retrieved aerosol compositions is then summarized.

## 3.7 MERRA2 apriori information

AOD can be derived from the product of aerosol number concentration and the particle extinction cross-section. For a given aerosol component, the column AOD at wavelength $\lambda$ can be written as

$$\text{AOD}(\lambda) = \int_0^\infty N(z)\sigma_{\text{ext}}(\lambda)\mathrm{d}z, \tag{18}$$

where $N(z)$ is the particle number concentration at height $z$, and $\sigma_{\text{ext}}(\lambda)$ is the extinction cross-section of an individual particle.

The column-integrated number concentration $N_{\text{col}}$ is the integration of the particle number concentration $N(z)$

$$N_{\text{col}} = \int_0^\infty N(z)\mathrm{d}z, \tag{19}$$

So the Eq. 18 can be rewritten as

$$\text{AOD}(\lambda) = N_{\text{col}}\sigma_{\text{ext}}(\lambda). \tag{20}$$

The extinction cross section can be further written as

$$\sigma_{\text{ext}}(\lambda) = Q_{\text{ext}}(\lambda)\pi r_{\text{eff}}^2, \tag{21}$$

where $Q_{\text{ext}}$ is the dimensionless extinction efficiency and $r_{\text{eff}}$ is the effective particle radius.

Combining these expressions yields the aerosol column number concentration:

$$N_{\text{col}} = \frac{\text{AOD}(\lambda)}{Q_{\text{ext}}(\lambda)\pi r_{\text{eff}}^2}. \tag{22}$$

For each aerosol component in MERRA-2, representative values of $r_{\text{eff}}$ are adopted following the aerosol size assumptions used in the GEOS model. Accumulation-mode aerosols such as sulfate and organic carbon typically have effective radii on the order of $r_{\text{eff}} \sim 0.3$–$0.4\ \mu\text{m}$, while other components exhibit component-dependent sizes. The corresponding extinction efficiencies $Q_{\text{ext}}(\lambda)$ are obtained from MOPSMAP simulations using consistent refractive indices and particle size assumptions. Finally, the estimated column number concentrations are normalized across aerosol components to obtain relative number concentration ratios. These ratios are used as prior information in the retrieval.

## 4 Results

### 4.1 Characterization of the Synthetic Aerosol Optical Database

Figure 2 shows the mean normalized AOD spectra simulated by MOPSMAP for five pure aerosol types: sea salt, sulfate, black carbon (BC), dust, and insoluble aerosols. Each case corresponds to an idealized scenario in which a single aerosol type

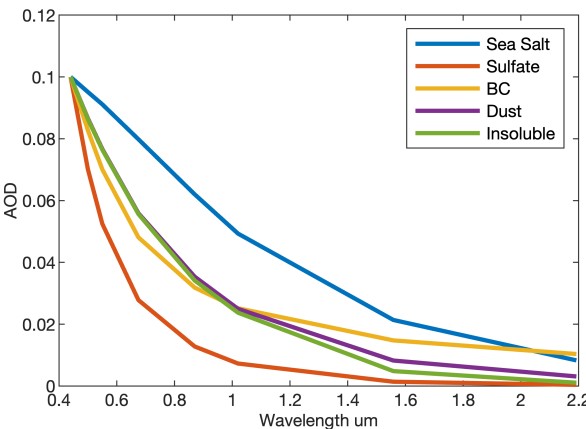

**Figure 2.** Normalized AOD spectra for five pure aerosol components simulated using MOPSMAP. Each spectrum assumes a single dominant aerosol type (100% composition) with AOD normalized at 440 nm.

dominates (100% composition), allowing a clear examination of spectral distinctions. The properties on the aerosol classes, e.g. complex refractive indices of different aerosol components, are based on the OPAC (Optical Properties of Aerosols and Clouds), which provides standard optical properties for atmospheric aerosols under diverse environmental conditions (Hess et al., 1998).

All spectra are normalized to an AOD of 0.1 at 440 nm. However, different aerosol types exhibit distinct spectral shapes, particularly in the shortwave infrared range. Sea salt shows the flattest spectral curve, maintaining relatively high AOD values across shortwave infrared wavelengths (e.g., 1.5–2.2μm), consistent with its coarse-mode size distribution and strong infrared extinction. In contrast, sulfate exhibits the steepest decline in AOD with wavelength, indicative of its fine-mode nature and low absorption.

Although BC, dust, and insoluble aerosols display very similar behavior in the visible range (440–870 nm), their differences become more distinguishable in the shortwave infrared. Dust retains slightly higher AOD values beyond 1.5μm due to its scattering efficiency at longer wavelengths, while BC and insoluble aerosols separate further in the 1.5 - 2 μm region.

These spectral distinctions arise from the different refractive indices and particle sizes of the aerosol types, and confirm the physical basis for using IR AOD as a complementary constraint in aerosol composition retrievals. Visible AOD alone may not

fully resolve composition degeneracies, especially between absorbing and scattering aerosols. Incorporating SWIR channels thus enhances the information content of the retrieval system, enabling improved discrimination of spectrally similar aerosol types. This supports the inclusion of SWIR AOD in our retrieval framework and highlights its practical relevance.

To better understand the diversity and coverage of the training dataset, we visualize the distributions of auxiliary parameters derived from the full synthetic database (all 10000 cases). Figure 3 shows the distributions of three key physical parameters

used as auxiliary inputs in the machine learning model: single scattering albedo (SSA), asymmetry factor (AF), and effective radius (Reff). These parameters are evaluated at 440 nm. The SSA histogram reveals a strong right-skewed distribution with a

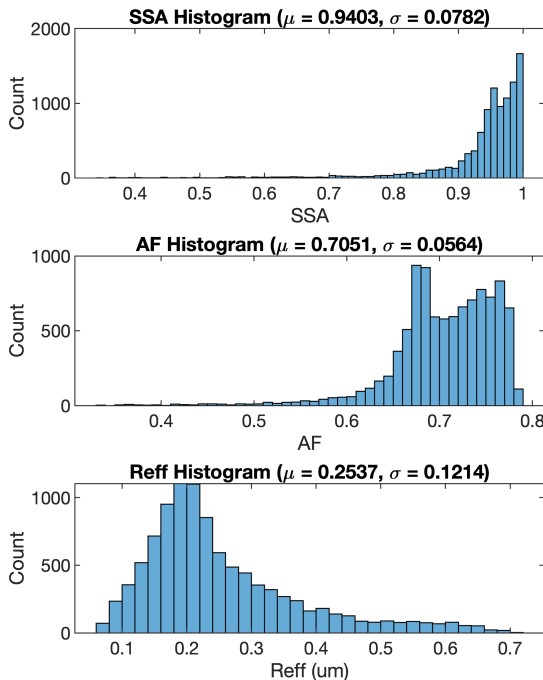

**Figure 3.** Histograms of key physical parameters in the training dataset: (top) Single scattering albedo (SSA), (middle) Asymmetry factor (AF), and (bottom) Effective radius (Reff), all evaluated at 440 nm. The mean and standard deviation of each parameter are indicated.

mean value of 0.94, suggesting most aerosols in the database are weakly absorbing. The AF histogram, centered around 0.70, reflects the forward-scattering nature of the aerosol mixtures. The Reff distribution peaks near 0.25 μm and spans from 0.1 to 0.7 μm, consistent with a mix of fine- and coarse-mode particles. These histograms demonstrate that the training database encompasses a wide range of realistic aerosol conditions.

### 4.2 Neural Network Trained Model vs. MOPSMAP

The machine learning model trained to replace MOPSMAP shows high accuracy in predicting AOD at multiple wavelengths. Figure 4(a-g) present a near-perfect agreement between predicted and original AOD, with $R^2$ values consistently above 0.99. The best performance is observed at 1.02 μm ($R^2 = 0.9964$), while all wavelengths exhibit minimal deviation from the 1:1 line, indicating reliable predictions of trained model.

The residual distribution in Fig.4(h) is centered around zero, confirming that prediction errors are symmetrically distributed with no systematic bias. The histogram shows that the majority of residuals remain within $\pm0.05$, further validating the model's precision.The performance metrics in the table highlight the robustness of the machine learning model. With a mean $R^2$ of

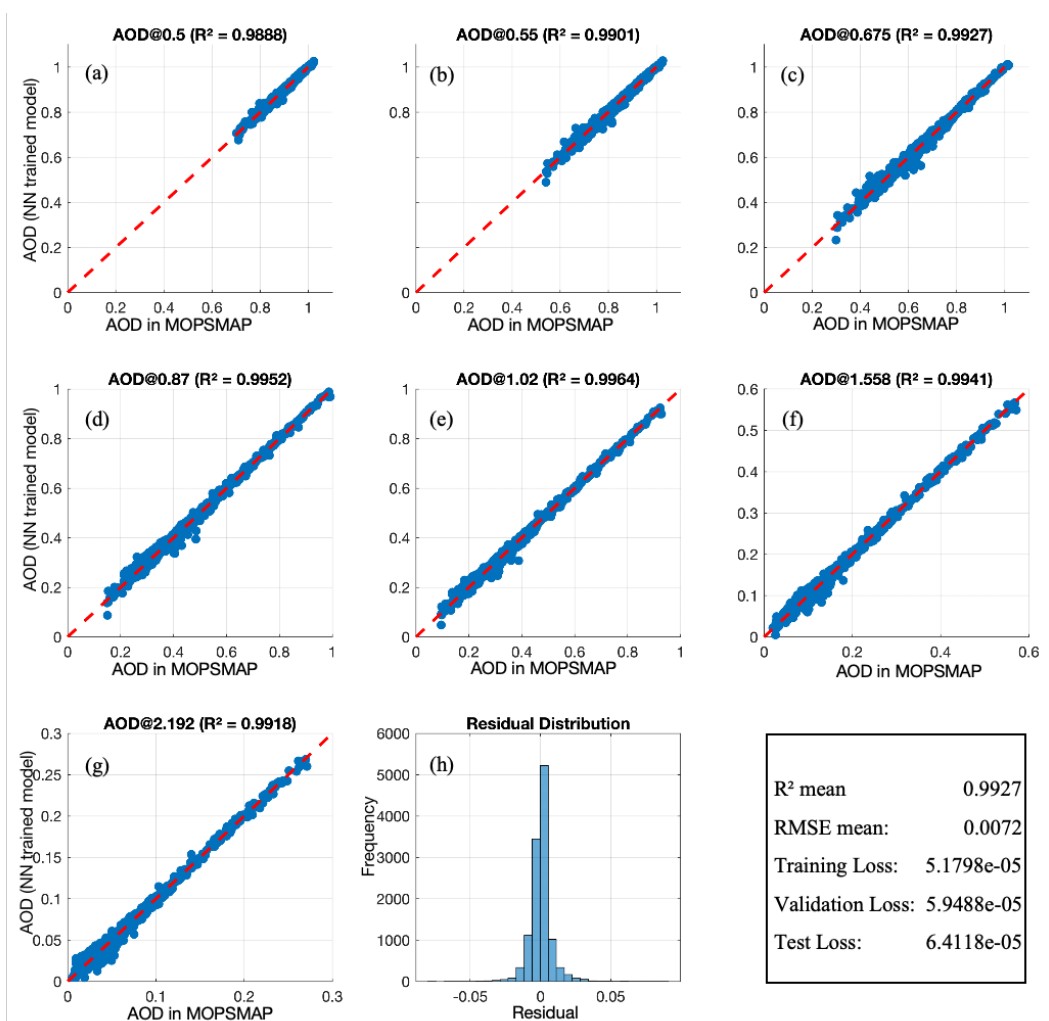

**Figure 4.** Performance evaluation of the machine learning model replacing MOPSMAP as the forward model. Figures (a-g) show scatter plots comparing the predicted normalized AOD from machine learning model with the normalized AOD from MOPSMAP at different wavelengths (0.5, 0.55, 0.675, 0.87, 1.02, 1.558, and 2.192 μm). The red dashed line represents the 1:1 reference line. Figure (h) displays the residual distribution of the predicted AOD values. The table summarizes key performance metrics, including the mean $R^2$, RMSE, and loss values for training, validation, and testing.

0.9927 and an RMSE of 0.0072, the model effectively captures the optical properties of aerosols. The low training, validation, and test losses (about $10^{-5}$) suggest strong generalization ability, minimizing the risk of overfitting.

Overall, these results confirm that the machine learning model successfully replicates the MOPSMAP simulations, offering an efficient and accurate alternative for forward modeling in aerosol retrieval.

**Table 2.** Normalized contributions (%) to the retrieval uncertainty for each aerosol component from prior, observation, and forward model error sources.

| Component | a Prior (%) | Observation (%) | Model (%) |
|---|---|---|---|
| Sea Salt | 53.8 | 39.4 | 6.8 |
| Sulfate | 57.6 | 36.0 | 6.4 |
| Black Carbon | 93.4 | 5.4 | 1.2 |
| Dust | 61.2 | 33.1 | 5.7 |
| Insoluble | 49.5 | 43.5 | 7.0 |

## 4.3  Retrieval Uncertainty Analysis

As we mentioned in Sec.3.5, to understand the source of retrieval uncertainty, we decompose the total posterior variance into contributions from prior, observation, and forward model errors. To better understand the relative importance of different uncertainty sources in the retrieval, we present their average contributions in Table 2, with further discussion below.

Table 2 summarizes the normalized contributions to the total retrieval uncertainty from a prior, observation, and forward model errors for each aerosol component, based on the 1500-case ensemble introduced in Section 3.5. The results highlight distinct sensitivities across aerosol types. For sea salt and sulfate, the a prior and observation contribute comparably (e.g., 53.8% vs. 39.4% for sea salt), indicating that these components are well constrained by the AOD spectral information. In contrast, black carbon retrieval is heavily dependent on a prior assumptions, with 93.4% of the uncertainty attributed to the a priori, reflecting its relatively weak spectral signature in the AOD spectrum. Dust and insoluble aerosols fall in between, with both a prior and observational constraints playing meaningful roles.

Importantly, the contribution from the forward model error remains below 10% for all aerosol types, confirming the stability and reliability of the machine-learning-based forward model used in this study. These findings underscore the benefits of combining physically consistent training datasets with efficient retrieval algorithms, enabling robust composition inference while keeping model-induced uncertainty low.

To quantify the information content of the retrieval, we compute the averaging kernel matrix $\mathbf{A}$. The diagonal elements of $\mathbf{A}$ reflect the sensitivity of each retrieved parameter to the observations. A value close to 1 indicates strong observational constraint, while values near 0 suggest the solution is mainly determined by the a prior. The averaged averaging kernel matrix obtained from 1500 synthetic test cases is:

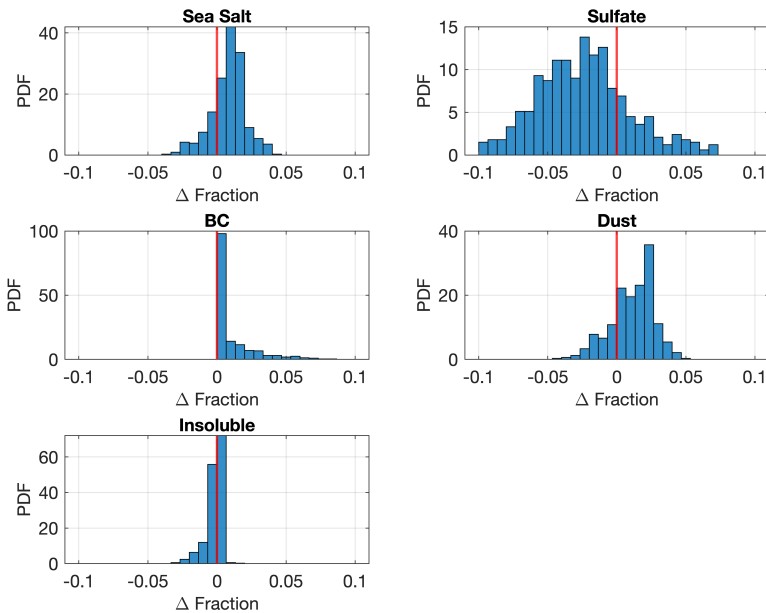

**Figure 5.** Distributions of relative changes in the retrieved aerosol component fractions from the sensitivity tests described in Section 3.6. Four auxiliary parameters (SSA$_{440}$, AF$_{440}$, Reff, RH) are perturbed, and deviations from the baseline retrieval are shown for each aerosol fraction. Red vertical lines indicate zero deviation.

$$\mathbf{A} = \begin{bmatrix} 0.85 & -0.00 & 0.15 & -0.01 & -0.03 \\ -0.00 & 0.89 & 0.18 & -0.05 & -0.03 \\ 0.15 & 0.18 & 0.27 & 0.23 & 0.15 \\ -0.01 & -0.05 & 0.23 & 0.86 & 0.00 \\ -0.03 & -0.03 & 0.15 & 0.00 & 0.83 \end{bmatrix}$$

The diagonal values indicate high sensitivity for sea salt ($A_{11} = 0.85$), sulfate ($A_{22} = 0.89$), dust ($A_{44} = 0.86$), and insoluble aerosols ($A_{55} = 0.89$), while black carbon is less constrained ($A_{33} = 0.27$). The total degrees of freedom for signal (DoF), given by $\mathrm{trace}(\mathbf{A})$, is 3.75, indicating that approximately 4 independent parameters can be resolved from the measurement.

## 4.4 Sensitivity analysis results

The distributions in Fig.5 show the deviations of the retrieved aerosol component fractions relative to the baseline retrieval obtained with fixed auxiliary parameters. For all aerosol types, the perturbation-induced deviations are narrowly distributed within $\pm 5\%$ and centered close to zero, indicating that uncertainty in the auxiliary parameters does not introduce systematic biases in the retrieval.

The dominant aerosol components (e.g., sulfate and dust) exhibit approximately symmetric distributions with typical spreads on the order of a few percent. In contrast, minor components such as black carbon and insoluble aerosol show strongly peaked distributions near zero, reflecting both their small absolute contributions and the limited sensitivity of the spectral observations to these components. More specifically, different aerosol components exhibit distinct responses to perturbations in the auxiliary parameters. Sea salt and dust tend to show slightly positive deviations, whereas sulfate exhibits a weak negative deviation. In contrast, black carbon and insoluble aerosol display no systematic offset, with their deviation distributions centered near zero.

It indicates that if uncertainties in the auxiliary parameters (SSA, asymmetry factor, effective radius, and relative humidity) are not explicitly accounted for in the inversion, the retrieval may underestimate sea salt and dust while overestimating sulfate. However, the magnitude of these deviations remains small, and the overall aerosol composition, particularly the dominant component, is not qualitatively affected.

## 4.5  Aerosol Composition Retrieval from Ground-Based Observations

Figure 6 presents the retrieved aerosol composition and corresponding AOD spectral fit at Ny-Ålesund on 21 April 2020. The retrieval results indicate that sea salt, sulfate, and black carbon aerosols dominate during this aerosol event. Specifically, sea salt constitutes the largest fraction (~43%), followed by black carbon (~33%) and sulfate (~24%). Dust and insoluble aerosols contribute minimally ($< 1\%$). The observed AOD spectrum is well constructed by the forward model (Fig. 6b), with residual differences typically below 0.005 (Fig. 6c).

To further assess the potential source regions of the retrieved aerosols, a 120-hour backward trajectory analysis is conducted using the HYSPLIT model (Fig. 6d). Based on the HYSPLIT back-trajectory analysis, the air masses (below 1500 m) are mainly originated from the ocean. Specifically, both 500 m and 1500 m trajectories indicate that, two days earlier (on 19 April), vertical lifting of air masses from the open ocean region between Canada and Greenland likely introduced sea salt aerosols into the lower troposphere, subsequently reaching Ny-Ålesund. Sea salt has been released in the atmosphere in the lowest 500 m between northeast Greenland on the last day prior to advection towards Ny-Ålesund. This transport pattern supports the presence of sea salt in the retrieved result. The upper-level trajectory (around 3000 m altitude) originates near the US-Canada border, suggesting sulfate and black carbon aerosols transported over longer distances (approximately five days) from anthropogenic sources in North America. These trajectories support the retrieved aerosol composition, confirming sea salt dominance from lower-altitude oceanic pathways and sulfate and black carbon from long-range transport at higher altitudes. Overall, this retrieval approach, integrating machine learning and optimal estimation, successfully captures aerosol composition with high accuracy and consistency between observed and modeled AOD spectra.

## 5  Conclusions

This study shows the feasibility of integrating machine learning with physically based aerosol modeling to retrieve aerosol composition from multi-wavelength AOD observations. By using a neural network trained on a comprehensive database generated with MOPSMAP, we successfully emulate the aerosol spectral features with aerosol composition and optical properties. The

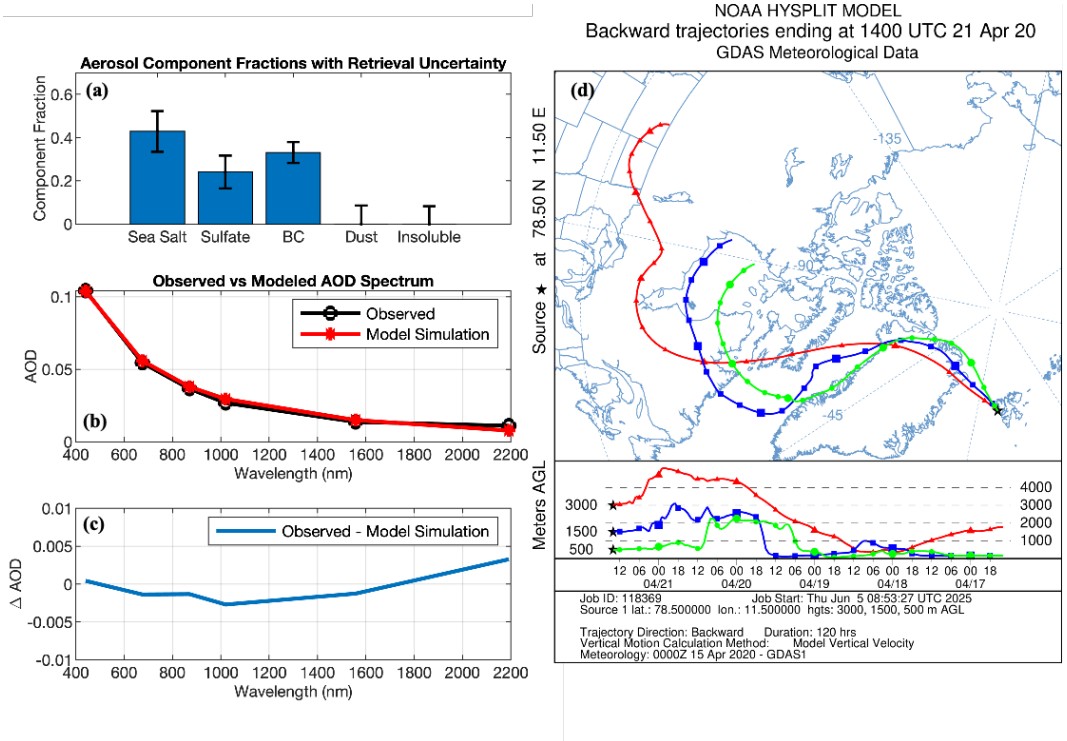

**Figure 6.** Aerosol composition retrieval from ground-based observations at Ny-Ålesund on 21 April 2020. (a) Retrieved aerosol component number concentration fractions (sea salt, sulfate, black carbon, dust, and insoluble aerosols) with uncertainties. (b) Observed versus modeled AOD spectra. (c) Residual differences between observed and simulated AOD. (d) 120-hour backward trajectories arriving at Ny-Ålesund (78.9°N, 11.9°E) at altitudes of 500 m, 1500 m, and 3000 m above ground level, computed using the NOAA HYSPLIT model (Stein et al., 2015).

resulting algorithm is easier to converge and efficient, suitable for application across diverse platforms, including ground-based FTIR and AERONET observations. In addition, this retrieval method is faster (about 5 - 10 times) than traditional full-physics
retrieval method, making it a promising tool for large-scale aerosol monitoring. The degrees of freedom for signal (DoF) analysis confirms the robustness of the retrieval framework. The diagonal elements of the averaging kernel matrix show strong observational constraints for sea salt, sulfate, dust, and insoluble aerosols ($A_{11} = 0.85$, $A_{22} = 0.89$, $A_{44} = 0.86$, $A_{55} = 0.89$), while black carbon is less constrained ($A_{33} = 0.27$), highlighting its stronger dependence on the prior. The total DoF of 3.75 suggests that approximately four independent aerosol parameters can be reliably retrieved from the multi-wavelength AOD
observations.

In this study, the use of Mie theory assumes spherical aerosol particles, which may introduce biases, particularly for nonspherical particles such as mineral dust. Determining aerosol shape is a complex problem that cannot be addressed solely through passive radiometry. In practice, characterizing particle asphericity requires additional measurements, such as polarization or depolarization ratios from lidar or radar. Our previous work (Ji et al., 2023) has demonstrated the feasibility of

combining FTIR and radar observations to jointly constrain aerosol properties. Therefore, incorporating aerosol shape information through active remote sensing is a promising avenue for future improvement of the retrieval framework.

Besides, our analysis highlights the importance of high-quality auxiliary optical parameters, particularly single scattering albedo (SSA), asymmetry factor (AF), and effective radius (Reff), in constraining aerosol composition retrievals. In ground-based settings, instruments such as AERONET offer reliable inversions of these parameters, enabling accurate component estimation when combined with spectrally resolved AOD. While satellite retrieval of these quantities remains more challenging. Current and upcoming satellite sensors can provide some of this information, but typically require multi-angle, multi-spectral, or polarimetric measurements, along with advanced inversion algorithms. For example, the POLDER instrument aboard PARASOL enabled global retrievals of SSA, Reff, and AF through polarization-based algorithms such as GRASP and RemoTAP (Dubovik et al., 2011b; Hasekamp and Landgraf, 2007). Upcoming missions like 3MI (on MetOp-SG) and NASA's PACE will carry advanced polarimetric imagers and hyperspectral sensors to further improve retrievals of aerosol microphysical and optical properties (Werdell et al., 2019; Gao et al., 2021a). However, these satellite-based parameters often originate from different sensors and require cross-platform coordination, unlike AERONET which provides all relevant quantities from a single instrument. Therefore, applying the proposed retrieval framework to satellite observations may involve higher uncertainty and greater dependence on auxiliary data. Future improvements in satellite instrumentation and algorithm synergy will help extend this framework from ground-based to global applications.

In summary, the integration of machine learning and multi-band AOD observations presents a promising method for aerosol composition retrieval. Continued efforts to improve shortwave infrared AOD accuracy, expand physical realism in training data, and incorporate additional observational constraints such as lidar profiles will be essential for achieving reliable, global-scale aerosol monitoring.

*Data availability.* The MERRA-2 reanalysis data used as a priori information are available from NASA's GES DISC at https://disc.gsfc.nasa.gov/datasets/M2T1NXAER_5.12.4 (hourly) and https://disc.gsfc.nasa.gov/datasets/M2TMNXAER_5.12.4 (monthly mean). Ground-based aerosol optical depth (AOD) observations are obtained from AERONET (https://aeronet.gsfc.nasa.gov/). The synthetic aerosol dataset used for model training and testing, as well as the code implementing the retrieval algorithm, are available upon request from the corresponding author. FTIR AOD observations are also available upon request from the corresponding author.

**Appendix A: Full-physical OEM inversion and analysis**

To quantify the fundamental limitations of retrieving aerosol microphysical parameters from spectral AOD and optical properties, we conducted a series of optimal estimation (OEM) experiments using a fully physical forward model based on MOPSMAP. The aim is to examine how a high-dimensional microphysical state vector behaves in an inversion framework and to show that the retrieval becomes effective only when the dimensionality of the state vector is reduced. These experiments also illustrate the limitations of purely physical forward models in representing the microphysics–optics mapping, motivating the use of machine-learning–based forward models.

## A1 Full-microphysical experiment

According to the MOPSMAP, we construct a 15-dimensional state vector,

$$\mathbf{x} = [r_1, \sigma_1, n_{0,1},\ r_2, \sigma_2, n_{0,2},\ r_3, \sigma_3, n_{0,3},\ r_4, \sigma_4, n_{0,4},\ r_5, \sigma_5, n_{0,5}],$$

representing five log-normal aerosol modes with mode radius $r$, width of the aerosol size distribution $\sigma$, and aerosol number concentration $n_0$.

The observation vector contains multi-wavelength AOD and optical quantities,

$$\mathbf{y} = [\mathrm{AOD}(\lambda_1), \ldots, \mathrm{AOD}(\lambda_8),\ \mathrm{SSA}_{440},\ R_{\mathrm{eff},440},\ g_{440},\ \mathrm{RH}],$$

a total of 12 observational elements.

The cost function is

$$J(\mathbf{x}) = (\mathbf{y} - \mathbf{F}(\mathbf{x}))^\top \mathbf{S_y}^{-1}(\mathbf{y} - \mathbf{F}(\mathbf{x})) + (\mathbf{x} - \mathbf{x_a})^\top \mathbf{S_a}^{-1}(\mathbf{x} - \mathbf{x_a}),$$

where $\mathbf{x_a}$ and $\mathbf{S_a}$ are the prior and prior covariance. Each iteration updates the state using the Gauss–Newton step,

$$\mathbf{x}_{i+1} = \mathbf{x}_i + \mathbf{G}\left[\mathbf{y} - \mathbf{F}(\mathbf{x}_i)\right], \quad \mathbf{G} = \left(\mathbf{K}^\top \mathbf{S_y}^{-1}\mathbf{K} + \mathbf{S_a}^{-1}\right)^{-1}\mathbf{K}^\top \mathbf{S_y}^{-1},$$

where $\mathbf{K} = \partial \mathbf{F}/\partial \mathbf{x}$ is the Jacobian.

In summary, the design of the OEM experiments presented above follows the same general framework as the method used in the main text. The key difference is that the forward model is the physical MOPSMAP model, which requires a 15-dimensional state vector.

The averaging kernel matrix,

$$\mathbf{A} = \left(\mathbf{K}^\top \mathbf{S_y}^{-1}\mathbf{K} + \mathbf{S_a}^{-1}\right)^{-1}\mathbf{K}^\top \mathbf{S_y}^{-1}\mathbf{K},$$

quantifies how much each retrieved parameter is constrained by the measurements.

Using MOPSMAP, an artificial observation is generated. Therefore, the "true" microphysical state $\mathbf{x}_{\mathrm{true}}$ is known. The results of this artificial observation show that, for the full 15-dimensional state vector, the degrees of freedom for signal (DOFS) is only about 1.3, and most diagonal elements of the averaging kernel matrix $\mathbf{A}$ are much smaller than 0.1. This means that only 1–2 independent combinations of microphysical parameters are actually constrained by the optical observations, while the remaining 13 dimensions are dominated by the apriori. In other words, the full microphysical parameter space is fundamentally under-determined with the available measurements, reflecting a strongly ill-posed retrieval problem.

## A2 Fixed aerosol size experiment

As we mentioned before, the observation in this experiment is generated by MOPSMAP, the true aerosol size parameters are known. This allows us to test whether fixing the aerosol size parameters can improve the inversion performance. Therefore, the

state vector is reduced to $\mathbf{x} = [n_{0,1}, \ldots, n_{0,5}]$. The corresponding average kernel function is:

$$\mathbf{A} = \begin{bmatrix} 0.68 & -0.27 & -0.07 & -0.18 & -0.16 \\ -0.27 & 0.55 & 0.02 & -0.29 & -0.01 \\ -0.07 & 0.02 & 0.01 & 0.02 & 0.01 \\ -0.18 & -0.29 & 0.02 & 0.51 & -0.06 \\ -0.16 & -0.01 & 0.01 & -0.06 & 0.21 \end{bmatrix}$$

The averaging kernel matrix shown above reveals how the simplified state vector responds to perturbations in the true aerosol composition. The diagonal elements indicate that approximately two aerosol components are substantially constrained by the observations, while the remaining modes retain partial or weak sensitivity. In contrast to the full 15-dimensional microphysical inversion, here the aerosol size parameters are fixed to their known values. This dimensionality reduction removes major degeneracies between size and composition parameters, leading to a significant increase in the effective information content.

With the size parameters fixed, the inversion focuses solely on the five aerosol number concentrations, allowing the AOD spectral measurements to constrain the aerosol composition more effectively. The near-unity diagonal values for several modes (0.68, 0.55, 0.51) demonstrate that the retrieval can successfully capture variations in these components. The resulting DOFS of 1.96 indicates that meaningful composition information emerges only when the state vector is reduced to match the intrinsic information content of the observations.

These two experiments show that the microphysics–optics mapping is highly non-unique. A full 15-dimensional microphysical inversion cannot be constrained by spectral AOD, SSA, AF, and Reff. Much of the microphysical complexity must remain in the forward model rather than the inversion state vector. Fixing the aerosol size parameters clearly improves the retrieval performance, suggesting that constraining additional observable optical properties, such as SSA and AF, would further improve the retrieval if such constraints could be imposed. However, in MOPSMAP these quantities are outputs of the model and therefore cannot be fixed directly. This limitation motivates the machine-learning–based forward model used in the main text, which accepts SSA, AF, and other bulk optical parameters as inputs and embeds the microphysical complexity within the learned mapping. The full-physical OEM experiments therefore justify the reduced parameterization adopted in our machine-learning–aided inversion framework.

*Author contributions.* J.D. conceived the study, developed the retrieval framework, conducted the data analysis, and led the manuscript writing. X.S. contributed to algorithm development and data interpretation. C.R. and J.N. reviewed the manuscript and provided feedback. All authors contributed to the final version of the paper.

*Competing interests.* Justus Notholt is a member of the editorial board of Atmospheric Measurement Techniques. The authors declare that they have no other competing interests.

*Acknowledgements.* We gratefully acknowledge the funding by the Deutsche Forschungsgemeinschaft (DFG, German Research Foundation) – project number 268020496 – TRR 172, within the Transregional Collaborative Research Center "ArctiC Amplification: Climate Relevant Atmospheric and SurfaCe Processes, and Feedback Mechanisms (AC)3". We thank the senate of Bremen for partial funding of this work.

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
