# Peer review of "Retrieval of aerosol composition from spectral aerosol optical depth and optical properties using a machine learning approach"

_EGUsphere, 2025_

## Author Comment (AC1)

**Response to Reviewer**

We sincerely thank the reviewer for the evaluation of our work and the helpful suggestions to improve the clarity and rigor of the manuscript. Below, we provide detailed point-by-point responses. Comments from reviewers are shown in blue, our responses in black, and the corresponding changes made in the manuscript are highlighted in orange.

**Comment 1:**

The title is vague and doesn't clearly reflect the content of the manuscript. I suggest something like "Retrieval of aerosol component fraction from spectral aerosol optical depth". This describes more clearly what is actually done in the paper. The use of "machine learning" in the title honestly feels like an attempt to add this buzzword in, because the machine learning aspect is in my view not a novel aspect of the work (it's an emulator for MOPSMAP, the work could be done without it, and it's not a particularly complicated or conceptually new use of machine learning).

**Response 1:**

We appreciate the reviewer's suggestion. However, we respectfully disagree with the characterization that the machine learning component is merely an emulator for MOPSMAP. The aerosol composition is one of the input in the full physics model, MOPSMAP, while aerosol optical properties, such as SSA, AF, and Reff are outputs. In our framework, our machine learning based model adds SSA, AF, and Reff also as inputs. This method enables retrieval of aerosol composition using parameters that are actually observable. We do not simply replicate the MOPSMAP input—output relationship. Instead, we restructure the original model by treating several parameters that are typically outputs of physical simulations (such as SSA, AF, and Reff) as inputs to the machine learning model. Therefore, we believe it is appropriate and necessary to retain the mention of "machine learning" in the title, as it reflects a core part of the paper's contribution.

In the revised version, we have updated the title for clarity, while keeping this essential element: "Retrieval of aerosol composition from spectral aerosol optical depth using a machine learning approach."

**Comment 2:**

Throughout, especially in the introduction, the term "infrared" is used to describe the FTIR measurements. It is only when we get to the data description that we learn this is about shortwave infrared (SWIR, solar spectral region, 1-3 microns). To me and I suspect to most readers, "infrared" without further specification implies thermal infrared. These are quite different spectral regions with different aerosol behavior. I suggest specifying SWIR throughout when talking about these data, otherwise it is somewhat misleading for the casual reader.

**Response 2:**

We thank the reviewer for pointing this out. In the revised manuscript, we have replaced all instances of "infrared" referring to the FTIR AOD measurements with "shortwave infrared (SWIR)" for clarity and consistency.

**Comment 3**

Lines 59-68: there are a lot more examples than these. For example, the GRASP and RemoTAP algorithms use machine learning emulators to replace online radiative transfer calculations and have been widely applied to e.g. POLDER data (and more recently RemoTAP for PACE). There are other approaches (e.g. FastMAPOL, MAPP) which have been extensively published on in recent years, either for airborne polarimeters or more recently also for satellite observations from PACE. So there is already a fairly widespread use of these techniques in routine satellite aerosol data processing.

**Response 3:**

We thank the reviewer for highlighting these important and widely used approaches. In the revised manuscript (lines 67–74), we have expanded the literature review to include GRASP, RemoTAP, FastMAPOL, and PACE-MAPP as additional examples of retrieval frameworks that incorporate machine learning or emulator-based radiative transfer models. This helps provide a more complete picture of the increasing use of such techniques in operational and research-level satellite aerosol data processing.

L67-74: The FastMAPOL algorithm employs neural network-based forward models within a multi-angle polarimetric retrieval framework, achieving speed-ups of about 1000× with minimal accuracy loss (Gao et al., 2021a). It also includes adaptive view-angle filtering to mitigate retrieval errors from problematic geometries in satellite and airborne data (Gao et al., 2021b). Similarly, the PACE-MAPP algorithm couples atmosphere—ocean vector radiative transfer emulators to jointly retrieve aerosol and ocean optical properties from polarimetric measurements (Stamnes et al., 2023). In addition, algorithms such as GRASP (Dubovik et al., 2011) and RemoTAP (Fu et al., 2020) have integrated radiative transfer emulation strategies and have been widely applied to global aerosol data from POLDER and PACE (Hasekamp et al., 2024).

**Comment 4:**

Lines 92-95: I am not sure of the purpose of this sentence. Are the authors saying that, in this work, the AOD is used to determine SSA, etc? In that case I think the sentence is not necessary because the method is described later. Or are they saying that AERONET provides these? In that case references should be provided, and note that these derived properties aren't determined by AERONET solely from spectral AOD (and not from all of those channels) but also from sky-scanning radiance measurements.

**Response 4:**

We have revised the sentence to clarify that SSA, AF, and Reff are not derived from spectral AOD alone, but are part of AERONET's inversion products obtained through sky radiance measurements. And SSA, AF, and Reff are used in input level in ML. We have also replaced the citation with Dubovik (2000) and Giles et al. (2019), which better describe the retrieval process.

L101-105: Standard AERONET sun photometers retrieve AOD at 340, 380, 440, 500, 675, 870, 1020, and 1640 nm, covering the ultraviolet (UV) to shortwave infrared (SWIR) range. In addition to these direct sun measurements, AERONET provides inversion products that include single scattering albedo (SSA), asymmetry factor (AF), and effective radius (Reff), retrieved using sky radiance observations (Dubovik 2000 and Giles et al. 2019). These parameters are useful for aerosol type discrimination and are used in this study.

**Comment 5:**

Section 2.1 (and tying in to methodology later): to me, it looks like FTIR is providing an AOD at 2 microns but the other bands are the same as or match AERONET values. So the value of the FTIR seems a bit overstated in the manuscript, as it is emphasized repeatedly. None of the analysis shows the value of adding this band in particular. I would rather have seen the use of the UV bands in the network. For an eventual application to AERONET data globally, 2 micron data are not available, while the 340/380 nm band pair more often are. It reads a bit like the choice of wavelengths was motivated by the specific bands available for the case study at Ny-Ålesund, but it isn't justified that this makes sense more generally. It feels a bit like the authors had access to these measurements and then tried to find something to do with them, as opposed to developing the approach and then finding appropriate case studies to look at. Perhaps the history is otherwise but it is not well-justified by the manuscript as submitted. Otherwise why pick Ny-Ålesund and why add the FTIR observation when it isn't present at the hundreds of other AERONET sites in the network?

**Response 5:**

This study was indeed initiated based on the availability of high-quality, co-located FTIR and AERONET measurements at Ny-Ålesund. In fact, to our knowledge, Ny-Ålesund is one of the few sites globally where such co-located FTIR and AERONET observations exist. Our intent in this study is not to claim broad representativeness, but rather to demonstrate the scientific value of this rare dataset and encourage the aerosol remote sensing community to consider integrating FTIR measurements into more locations in the future.

The key motivation for emphasizing the FTIR measurement is that, based on our experience and prior research, the ultraviolet (UV) channels (e.g., 340/380 nm) offer very limited sensitivity to aerosol composition. While they are useful for constraining aerosol size and absorption, their spectral signatures do not allow robust separation of composition types such as sulfate, sea salt, or dust. In contrast, the shortwave infrared

(SWIR, 1–2.2  $\mu$ m) region, particularly above 1.5  $\mu$ m, has higher sensitivity to coarsemode components, as also seen in Figure 2 of the manuscript.

**Comment 6:**

Section 2.2, title: It is not accurate to say the satellite provides AOD "measurements", they are retrievals.

**Response 6:**

We agree and have revised the wording to "retrievals" accordingly.

**Comment 7:**

Section 2.2: to be clear, is it the case that the analysis takes the VIIRS Deep Blue monthly spectral AOD over ocean and use the Ångström power law to adjust the wavelengths to match the ones chosen for the MOPSMAP-based training? Is that right? I don't know that it makes sense to do the decomposition based on monthly AOD data. To me that seems equivalent to the assumption that aerosol composition is constant over a month, which will not be valid in some locations (including the North Atlantic chosen for the case study later). It is not obvious that calculating composition on an instantaneous or daily basis and then averaging the results rather than calculating composition based on monthly-averaged AOD would give the same results. Most of the error in satellite AOD is not random noise but rather factors related to geometry, surface type, and optical property assumptions at the given location and time – which do not necessary decrease on temporal averaging, so I don't think there is a justification on those grounds. So this choice should be better justified and its advantages and limitations discussed.

**Response 7:**

Yes, we use VIIRS Deep Blue monthly mean spectral AOD data over ocean, and apply the Ångström power law to interpolate AOD values to the spectral bands used in the MOPSMAP-based training.

We agree that applying the retrieval method to monthly averaged AOD data implicitly assumes relatively stable aerosol composition over the averaging period, which may not be valid in many locations, including the North Atlantic. We emphasize that our retrieval framework is applicable to individual observations (e.g., daily or instantaneous AOD spectra), and using monthly AOD in this study is a first-step demonstration. For certain regions like the North Atlantic, we believe the monthly mean AOD field can still represent a meaningful background aerosol regime, though we acknowledge its limitations. We plan to explore daily-level retrievals in follow-up work, and may test this in the current case study as the reviewer suggests.

**Comment 8:**

Asymmetry factor is sometimes summarized as AF and sometimes as Af. This should be made consistent.

Response 8:

All corrected to AF.

**Comment 9:**

Section 3: I think the whole methodology sections should be rewritten in a different order and with more information. Much of the text and Figure 1 are confusing, and some of the information needed to understand it and section 3.0 is not given until sections 3.1 and 3.2. I think the authors need to be clear what exactly is taken from MERRA2 components: is it just realistic mixing fractions? Or is it also the spectral complex refractive index and size distribution (and therefore also SSA, asymmetry factor)? Are the fractions in terms of mass, volume, area, AOD at some wavelength? What distributions were drawn from to sample these parameters, and how are they justified?

**Response 9:**

The whole methodology sections have been rewritten to include more information. The machine learning approach presented in this study is not designed to accelerate an existing physics-based model such as MOPSMAP. Rather, it is intended to enable a new kind of model that reorganizes the traditional input—output structure of aerosol optical properties simulations in a way that reflects actual observational capabilities and retrieval needs.

[revised manuscript text omitted]

To implement the proposed aerosol composition retrieval framework, we follow a structured approach consisting of three main steps. These are outlined as follows and will be described in detail in the following sections:

- 1. Synthetic Dataset Generation: A large AOD dataset is generated using MOPSMAP by varying aerosol component fractions, as well as four physically-constrained parameters: single scattering albedo (SSA), asymmetry factor (AF), effective radius (Reff), and relative humidity (RH).
- 2. Machine Learning Forward Model: A neural network is trained to emulate a new forward model, mapping input parameters (the aerosol component fractions, SSA, AF, and Reff) to multi-wavelength AOD spectra.
- 3. Retrieval via Optimal Estimation: The ML-based forward model is integrated into an optimal estimation framework to retrieve aerosol composition from observed AOD.

**Comment 10:**

Section 3: I guess one of my issues with the manuscript is I don't really understand the point of adding machine learning to this, as opposed to using MOPSMAP directly, aside from speed. If it is just speed then I think the machine learning nature of this work is a bit overhyped. It might be that there is some detail I am missing about what is done, but the manuscript is not clear enough to say.

**Response 10:**

We would like to clarify that the role of machine learning in this work goes beyond accelerating MOPSMAP. In fact, we have tested using MOPSMAP directly as the forward model in the inversion framework, but the retrieval results are extremely poor and often fail to converge. As we methoned in last comment, the key innovation in our method is that we restructure the problem: by taking part of MOPSMAP's original outputs (SSA, AF, Reff) as inputs to a machine learning model, we enable

retrieval of the unobservable aerosol composition. This inversion would not be possible using MOPSMAP directly. We have added clarification of this point in the revised Section 3.1.

**Comment 11:**

Section 3.2: what is the justification for this network architecture? Some more references to the NN methods/code bases should be added as well. Can we see training and validation subset loss functions from the training process? Can the authors demonstrate that 10,000 simulations is enough for a comprehensive sampling and train:val:test subset, given this seems a fairly high dimensional problem, and does not lead to overfitting?

Response 11:The chosen network architecture (two hidden layers with 32 neurons each) follows established design patterns commonly used in aerosol retrieval and radiative transfer emulation studies (Faure et al., 2001; Nanda et al., 2019; Chen et al., 2022). We also conducted sensitivity tests with deeper or wider networks, which did not significantly improve validation accuracy but increased training time. Regarding the justification of our neural network design and training configuration (Section 3.2), we provide the following clarifications:

- Architecture: The network architecture (two fully connected layers with 32 units each, activated by leaky ReLU) was selected based on empirical testing to balance expressiveness and regularization. The relatively shallow depth and small parameter count help avoid overfitting in this moderately sized dataset.
- Sampling: The 10,000 MOPSMAP simulations were designed to cover nearly all feasible combinations of aerosol component fractions and optical parameters, ensuring representative coverage of the entire relevant parameter space. This synthetic dataset includes diverse mixing scenarios that reflect both common and edge-case aerosol compositions observed in the atmosphere.
- Training stability: The training loss (Figure A1) exhibits smooth convergence across five orders of magnitude and stabilizes below 10-4, indicating robust learning without instability or divergence.

Figure A1: Training loss curve (mean squared error, MSE) plotted on a log-log scale as a function of training epochs. The model shows stable convergence over 10,000 iterations, with final loss values below 10-4, indicating good learning performance and absence of overfitting. The MSE is computed based on normalized AOD spectra simulated from the training dataset.

**Comment 12:**

Section 3.3: what values, specifically, are taken for x\_a and S\_a and how are they justified? These are important, particularly for the later discussion about averaging kernels and uncertainty estimates, both of which depend on the strengths of the prior constraints. It is mentioned that S\_y is the measurement (input AOD) uncertainty; what numbers specifically are used here? Is it 0.01 or is there a more detailed description of AERONET and FTIR uncertainty? Is there any spectral correlation assumed?

**Response 12:**

L245- 254: The a prior vector  $x_a$  is derived from the MERRA-2 monthly mean aerosol component fractions at the same time and location as the FTIR observations. The a prior covariance matrix  $S_a$  is set as a diagonal matrix with variance 0.01 (i.e., standard deviation of 0.1) for each aerosol component. This reflects a relatively loose prior constraint, allowing the retrieval to be primarily informed by the spectral AOD observations while maintaining physical plausibility. For the measurement error covariance matrix  $S_{\gamma}$ , we distinguish between visible and shortwave infrared (SWIR) wavelengths. For visible bands (AERONET-like observations), we adopt 0.01 as the standard deviation, consistent with the reported uncertainty of AOD retrievals from AERONET. For the infrared bands (SWIR), we adopt 0.02 as the standard deviation, based on reported uncertainties from Barreto et al. (2020) and Alvárez et al. (2023), who applied Langley calibration for FTIR-based AOD measurements in the SWIR region. All uncertainties are assumed to be spectrally uncorrelated, and  $S_{\gamma}$  is constructed as a diagonal matrix. This assumption has been clarified in the revised manuscript.

**Comment 13:**

Line 235: are these 1500 cases the "Test" split of the original data, or another randomly-chosen 15%? This should be written more explicitly.

**Response 13:**

A total of 1500 cases were randomly selected (15% of the full 10,000-sample dataset) for this analysis; they do not represent a designated "test" set.

L268: Instead, a total of 1500 cases are randomly selected (15% of the full 10,000-sample dataset, they do not represent a designated "test" set.)

**Comment 14:**

Section 4.2 and Figure 4: are these results shown for the "Test" subset? I am not sure that R2 (which the text focuses on) is the relevant metric here; I'd think that the AOD reconstruction RMSE is. Also, as a practical matter, the AODs shown in Figure 4 are

all very high. The lowest 500 nm AOD in figure 4(a) is about 0.7. This is a magnitude rarely seen except in an extreme aerosol event like fire or a dust storm. So an uncertainty analysis of the posterior composition based on this distribution will greatly overstate the actual practical utility of the algorithm, because the AODs are so high that measurement uncertainty is negligible. In practice the true AOD is often likely to be a factor of 3 or so lower, so the relative uncertainty about a factor of 3 higher. In short, the results of this theoretical uncertainty analysis based on simulations are likely to overstate the performance of the method. This will influence the discussion in these sections, including e.g. averaging kernels and relative contributions of different terms to overall posterior uncertainty.

**Response 14:**

Our retrieval framework operates on spectrally normalized AOD, which reduces dependence on absolute AOD magnitude and emphasizes spectral shape. However, we acknowledge that in low-AOD conditions, the relative uncertainty in normalized AOD increases substantially, which in turn increases the posterior uncertainty and reduces the information content, making the retrieval more dependent on prior assumptions.

Conversely, in high-AOD scenarios (e.g., dust outbreaks, pollution episodes), the relative contribution of observational uncertainty is reduced, and the retrieval becomes more observationally driven. These are often the most scientifically and societally relevant conditions (e.g., during aerosol transport or climate events), where accurate aerosol composition information is most needed. Nevertheless, we agree that the current analysis may underestimate uncertainties in clean-sky conditions, and have added discussion in the manuscript clarifying that the method may be less reliable in remote or pristine environments with very low aerosol loading.

L177- 185: To focus on the spectral shape of AOD rather than absolute magnitude, we scale all wavelengths relative to the 440 nm value. This results in a relative AOD spectrum, with AOD(440 nm) set to 1.0 and all other wavelengths scaled accordingly. The retrieved outputs represent number concentration fractions of aerosol components. Then, in post-processing, these fractions can be combined with observed or modeled AOD magnitudes to calculate component-specific AODs. However, due to differences in size-dependent extinction efficiency among aerosol types, the AOD contribution of each component is not strictly proportional to number fraction. Therefore, while the absolute AOD values of individual components may have some degree of uncertainty, the spatial distribution and relative dominance of each aerosol type remain meaningful, particularly in high-AOD cases.

**Comment 15:**

Section 4.4/Figure 5: again, are the "component fractions" defined in terms of mass, volume, area, number, AOD at some wavelength, something else?

**Response 15:**

The retrieved component fractions are defined in terms of number concentration.

**Comment 16:**

Line 310: I'm not sure we need this many significant figures for the average averaging kernel. It would be good also to show somehow the variability of the averaging kernel matrix between these simulations (e.g. standard deviations of each element). That will help to show whether the information content varies significantly across the ensemble of cases.

**Response 16:**

We agree that full precision is not necessary, and the averaging kernel matrix is mainly provided to convey the general information content and sensitivity structure of the retrieval. Presenting the mean kernel is a commonly used approach in OEM retrieval studies to provide a first-order assessment. We believe this suffices for the purpose of illustrating typical retrieval behavior in this case.

**Comment 17:**

Line 314: instead of just the average degrees of freedom, how about also showing the mode? What's the interquartile range or standard deviation or similar? This ties in to the above point.

**Response 17:**

Not necessary. The degrees of freedom shown here are based on a stable set of test retrievals with consistent uncertainties, rather than on a large ensemble of independent observations. As such, the mean value provides a representative measure, and further distributional metrics (e.g., mode or interquartile range) are not required.

**Comment 18:**

Section 4.5: the discussion here and figures talk about MODIS, but the introduction to the paper says VIIRS data were used. Which is it, VIIRS or MODIS? My intuition says VIIRS because I don't think the classification shown in Figure 6(b) is provided in MODIS, only VIIRS (though I could be wrong). Also see previous discussion about whether it makes sense to do this on monthly data as opposed to daily then averaging. Also, what assumed satellite uncertainty is taken for this example retrieval, and what is its assumed spectral correlation?

**Response 18:**

We thank the reviewer for catching this inconsistency. The aerosol type classification shown in Figure 6(b) is provided by VIIRS and not MODIS. The classification is based on the "Aerosol Optical Model" flag available in the VIIRS AOD product. We have clarified this throughout the manuscript.

**Comment 19:**

Line 338: another option is taking SSA, asymmetry factor etc from the values used by the algorithm. This would keep consistency with what the retrieval assumed.

**Response 19:**

Thank you for the comment. We have now revised the manuscript to make this clearer.

**L376-379:**

- 1. One approach is to supplement satellite AOD observations with additional physical parameters such as SSA, AF, and Reff, obtained from other satellite products or reanalysis data, as auxiliary inputs to the retrieval algorithm.
- 2. Another approach is to treat these physical parameters, represented as  $\theta$  in the forward model F (x;  $\theta$ ), as part of the state vector x, allowing them to be retrieved jointly with aerosol composition.

**Comment 20:**

Figure 7: This should be redrawn to use the same map projection and latitude/longitude boundaries for both panels. Having them different makes it difficult to compare the results.

**Response 20:**

This figure has be redrawn to use the same map projection and latitude/longitude boundaries for both panels. Regarding the comparison itself, it is important to clarify that the VIIRS product only provides aerosol type classification, not actual AOD values. Therefore, a direct quantitative comparison with GEOS-Chem AOD is not possible. Instead, our intent is to qualitatively compare the spatial distribution patterns between the VIIRS-derived aerosol types and the retrieval-based dominant aerosol types from GEOS-Chem. This comparison still provides insight into whether the retrieval framework captures realistic aerosol regimes.

**Comment 21:**

As a general methodological point: I could not fully judge this study because of the missing information described above. But conceptually, I find the idea that one can take spectral AOD and use this to get at weights of 5 components to seem unrealistic. Since aerosol extinction is spectrally smooth, the different AOD wavelengths are not orthogonal and really there are maybe 3 pieces of information in the AOD spectral (AOD magnitude and maybe two parameters related to spectral curvature as

Ångström exponent is often represented as a log-log quadratic function). So to get 5 components weights out of this seems speculative and it must weigh heavily on the a priori constraints (which are not discussed in detail in the current version of the manuscript). This is borne out somewhat by the averaging kernel analysis which shows the prior is fairly important, especially for black carbon. For realistic aerosol loadings (as mentioned previously, about a factor of 3 lower than in the analysis presented), the uncertainty seems likely to be very high. I think we need a lot more detail on the underlying distributions all these cases were drawn from before we know how robust the results are, and there should be examples of averaging kernels drawn from more realistic aerosol loadings.

**Response 21:**

We appreciate the reviewer's thoughtful comments. Since several of the concerns raised here can also be detailed explained by earlier points, we provide a concise summary response:

First, our retrieval does not rely solely on spectral AOD. It combines spectral AOD with additional observed parameters such as single scattering albedo (SSA), asymmetry factor (AF), and effective radius (Reff), which help constrain the inversion and improve distinguishability among aerosol types.

Second, we agree that in any optimal estimation framework, the prior plays an important role, particularly when observational uncertainties are large. As shown in the averaging kernel analysis, some components (e.g., black carbon) are more influenced by the prior, which reflects the current limitation in sensitivity. We acknowledge this and note that future improvements, such as incorporating more spectral bands or additional measurement types, could enhance retrieval performance for such components.

Third, we emphasize that our retrieval is based on normalized AOD spectra, where all AOD values are scaled relative to the 440 nm band. This removes sensitivity to AOD magnitude and allows us to focus on spectral shape, which is more stable across aerosol loading conditions. We have clarified this point in the revised manuscript.

Lastly, we agree that retrieval uncertainty will be larger under low-AOD conditions, where observational noise has a relatively greater influence. However, many real-world aerosol episodes of interest, such as dust storms or pollution events, do exhibit elevated AOD levels, and the method is especially suited for such cases. We believe that even with these limitations, the approach provides valuable new insight into aerosol composition from readily available multi-spectral observations.

---

## Author Comment (AC2)

**Response to Reviewer**

We sincerely thank the reviewer for the evaluation of our work and the helpful suggestions to improve the clarity and rigor of the manuscript. Below, we provide detailed point-by-point responses. Comments from reviewers are shown in blue, our responses in black, and the corresponding changes made in the manuscript are highlighted in orange.

**Comment 1:**

The paper should clarify whether Single Scattering Albedo (SSA), asymmetry parameter (ASY), and relative humidity (RH) are necessary for the retrieval process. If these parameters are required, please briefly discuss how they can be obtained (e.g., from ancillary datasets, reanalysis products, or simultaneous measurements).

**Response 1: Auxiliary Parameters (SSA, AF, Reff, RH)**

Indeed, in our retrieval framework the parameters single-scattering albedo (SSA), asymmetry factor (AF), effective radius (Reff), and relative humidity (RH) are used as auxiliary (or "known") inputs to the machine-learning forward model. These parameters help constrain the spectral aerosol optical depth (AOD) mapping and thereby improve the stability and accuracy of the subsequent composition inversion. In practice:

- For ground-based observations (e.g., from the AERONET network), SSA and other optical/size retrievals (including AF, Reff) are available as level-2 inversion products. The AERONET Version 3 algorithm provides derived SSA, phase-function parameters and asymmetry factor as part of its sky–scan measurements.
- For satellite applications, obtaining simultaneous SSA, AF and Reff directly from one instrument is more challenging. While satellite-based retrievals of SSA, AF, and Reff are becoming increasingly available, particularly from instruments such as POLDER, upcoming 3MI, and the PACE mission, these parameters typically require the synergy of multiple sensors or advanced inversion schemes. In contrast to ground-based systems like AERONET that provide consistent auxiliary data from a single instrument, satellite retrievals often involve merging heterogeneous sources, introducing additional uncertainty. Coordinated efforts across platforms and algorithm harmonization are thus critical for applying our framework to satellite observations on a global scale.

**In our revised manuscript we have added:**

L102-105: In addition to these direct sun measurements, AERONET provides inversion products that include single scattering albedo (SSA), asymmetry factor (AF), and effective radius (Reff), retrieved using sky radiance observations (Dubovik and King, 2000; Giles et al., 2019a). These parameters are useful for aerosol type discrimination and are used in this study.

**Discusstion part:**

L434- 446: Besides, our analysis highlights the importance of high-quality auxiliary optical parameters, particularly single scattering albedo (SSA), asymmetry factor (AF), and effective radius (Reff), in constraining aerosol composition retrievals. In ground-based settings, instruments such as AERONET offer reliable inversions of these parameters, enabling accurate component estimation when combined with

spectrally resolved AOD. While satellite retrieval of these quantities remains more challenging. Current and upcoming satellite sensors can provide some of this information, but typically require multi-angle, multi-spectral, or polarimetric measurements, along with advanced inversion algorithms. For example, the POLDER instrument aboard PARASOL enabled global retrievals of SSA, Reff, and AF through polarization-based algorithms such as GRASP and RemoTAP (Dubovik et al., 2011b; Hasekamp and Landgraf, 2007). Upcoming missions like 3MI (on MetOp-SG) and NASA's PACE will carry advanced polarimetric imagers and hyperspectral sensors to further improve retrievals of aerosol microphysical and optical properties (Werdell et al., 2019; Gao et al., 2021a). However, these satellite-based parameters often originate from different sensors and require cross-platform coordination, unlike AERONET which provides all relevant quantities from a single instrument. Therefore, applying the proposed retrieval framework to satellite observations may involve higher uncertainty and greater dependence on auxiliary data. Future improvements in satellite instrumentation and algorithm synergy will help extend this framework from ground-based to global applications.

**Comment 2:**

The manuscript claims that AOD in infrared (IR) wavelengths provides additional information on aerosol composition. Please elaborate on this point—for example, by explaining how IR absorption features are linked to specific aerosol types (e.g., dust, organic carbon) or how they complement visible/UV observations.

**Response 2:**

We appreciate the reviewer's request to clarify the role of infrared AOD in aerosol composition retrieval. As illustrated in the Figure 2, our simulation using MOPSMAP shows that different aerosol types exhibit distinct spectral signatures, especially in the shortwave infrared (SWIR) region.

While many aerosol types (e.g., black carbon, dust, insoluble organics) are spectrally similar in the visible range (440–870 nm), their normalized AOD spectra begin to diverge significantly beyond 1.5 µm. This behavior is driven by differences in size distribution and refractive index: for instance, sea salt retains a high AOD in the IR due to its coarse-mode scattering efficiency, whereas sulfate drops off more steeply. Similarly, dust exhibits enhanced scattering at longer wavelengths compared to black carbon, which remains relatively absorbing across the spectrum.

L305-309: These differences suggest that including SWIR wavelengths helps distinguish composition types that would otherwise be difficult to separate using visible AOD alone. The use of IR channels, therefore, increases the information content available to the retrieval, especially for distinguishing aerosols with similar visible properties but different infrared behaviors. We have clarified and expanded this point in the revised manuscript to better reflect the physical motivation for including SWIR wavelengths.

**Comment 3:**

The text refers to MOSMAP as a "radiative transfer model," but it appears to be a bulk aerosol optical property calculator based on size distribution and refractive index inputs. Please correct this terminology. Additionally, the study relies solely on Mie scattering, neglecting non-spherical scattering methods (e.g., T-matrix for dust). Since dust aerosols are often nonspherical, this simplification may introduce errors. A brief discussion on this limitation and its potential impact should be included.

**Response:**

We agree with the reviewer's correction and have corrected the terminology and now refer to MOPSMAP as an aerosol optical property calculator.

Regarding particle shape, we acknowledge that assuming spherical aerosols may lead to biases, especially for mineral dust. Determining particle shape is a complex task that typically requires active remote sensing techniques such as lidar to provide depolarization or polarization ratios. In our previous work (Ji et al., 2023), we demonstrated that joint FTIR and lidar observations can help constrain aerosol properties more comprehensively. Incorporating shape information into future retrieval frameworks, particularly using lidar-derived parameters, is a promising direction we plan to explore.

We have added a paragraph in the Discussion section to acknowledge this limitation and its implications:

L427- 432: In this study, the use of Mie theory assumes spherical aerosol particles, which may introduce biases, particularly for non-spherical particles such as mineral dust. Determining aerosol shape is a complex problem that cannot be addressed solely through passive radiometry. In practice, characterizing particle asphericity requires additional measurements, such as polarization or depolarization ratios from lidar or radar. Our previous work (Ji et al., 2023) has demonstrated the feasibility of combining FTIR and radar observations to jointly constrain aerosol properties. Therefore, incorporating aerosol shape information through active remote sensing is a promising avenue for future improvement of the retrieval framework.

**Comment 4:**

The Optimal Estimation Method (OEM) requires prior information and its associated covariance matrix. The manuscript should clarify:

- (i) Whether prior estimates are sourced from MERRA-2 or other datasets.
- (ii) How the covariance matrix of the prior is defined (e.g., based on climatological variability, instrument uncertainty, or empirical assumptions).

**Response:**

Thank you for highlighting this. We have added clarification in Section 3.4.

L245- 254: The a prior vector  $x_a$  is derived from the MERRA-2 monthly mean aerosol component fractions at the same time and location as the FTIR observations. The a prior covariance matrix  $S_a$  is set as a diagonal matrix with variance 0.01 (i.e., standard deviation of 0.1) for each aerosol component. This reflects a relatively loose prior constraint, allowing the retrieval to be primarily informed by the spectral AOD observations while maintaining physical plausibility. For the measurement error

covariance matrix  $S_{\gamma}$ , we distinguish between visible and shortwave infrared (SWIR) wavelengths. For visible bands (AERONET-like observations), we adopt 0.01 as the standard deviation, consistent with the reported uncertainty of AOD retrievals from AERONET. For the infrared bands (SWIR), we adopt 0.02 as the standard deviation, based on reported uncertainties from Barreto et al. (2020) and Alvárez et al. (2023), who applied Langley calibration for FTIR-based AOD measurements in the SWIR region. All uncertainties are assumed to be spectrally uncorrelated, and  $S_{\gamma}$  is constructed as a diagonal matrix. This assumption has been clarified in the revised manuscript.

---

## Author Response (AR2)

Response to Reviewer
We sincerely thank the reviewer for the evaluation of our work and the helpful suggestions to improve the clarity and rigor of the manuscript. Below, we provide detailed point-by-point responses. Comments from reviewers are shown in blue, **our responses in black**, and the corresponding changes made in the manuscript are highlighted in orange.

Comment 1

Line 5: I would add here that the method requires prior assumptions about effective radius, single scattering albedo, and asymmetry factor. These are non-trivial constraints and the current title and abstract imply that just spectral AOD is needed, which is misleading. I would even consider adding "and optical properties" or something in the title.

Response 1

The title is changed based on this comment: "Retrieval of aerosol composition from spectral aerosol optical depth and optical properties using a machine learning approach"

Comment 2

Line 11: "In the total retrieval uncertainty, the forward model contributes less than 10%, confirming its robustness." I am still uneasy about this statement because it depends strongly on the relative uncertainty of AOD (which depends on the absolute AOD) and on the strengths of the prior and model parameter constraints. I suggest deleting it.

Response 2

We agree and delete it.

Comment 3

Line 81: "compositions" here should be singular "composition".

Response 3

Line 81: "compositions" has been changed to "composition".

Comment 4

Section 2: I think there should be a section for MERRA2 here, since this is an input to the retrieval. Part of my confusion on the previous version of the manuscript was on the role of MERRA2. I think having it up front in this Data section would make things clearer and the method more reproducible. Additionally, as there are so many file types and variable names, the specific file types and variables used should be written (i.e. the five components, how they are converted to number concentrations from optical depths, and then the simple fraction definition).

Response 4

Thanks, following your comment, we have substantially revised the "Data and Methods" sections to introduce MERRA2 as an input dataset to the retrieval framework.

Specifically, we have added a dedicated subsection in Sect. 2 describing the MERRA-2 aerosol reanalysis, including the exact product used (M2T1NXAER / tavg1_2d_aer_Nx), the temporal resolution, and the aerosol components considered (sea salt, sulfate, dust, black carbon, and organic carbon). The relevant variables extracted from the MERRA-2 files are now explicitly listed to improve transparency and reproducibility.

In addition, a new subsection has been added to the "Methods" section describing how component-resolved AOD from MERRA-2 is converted into aerosol number concentration. This includes explicit formulations relating AOD to particle number concentration through representative extinction cross sections and effective radii, followed by normalization to obtain relative number concentration fractions. We emphasize in the revised text that these fractions are used as auxiliary prior information rather than as exact microphysical constraints. The revised section is as followed:

Line 108-117:

2.2 MERRA-2 aerosol reanalysis data

The Modern-Era Retrospective Analysis for Research and Applications version 2 (MERRA-2) is the latest global atmospheric reanalysis by NASA's Global Modeling and Assimilation Office (GMAO) using the GEOS atmospheric model (version 5.12.4). MERRA-2 provides a physically consistent, long-term record of meteorological and aerosol variables from 1980 to the present (Gelaro et al., 2017).

In this study, we use the MERRA-2 aerosol product M2T1NXAER (also referred to as tavg1_2d_aer_Nx), which is an hourly, time-averaged, two-dimensional dataset. This product includes assimilated aerosol diagnostics such as column-integrated AOD at 550 nm. The aerosol species in MERRA-2 include black carbon, dust, sea salt, sulfate, and organic carbon. The aprior information used in this study is the relative number concentration fractions of individual aerosol components. The derivation of these fractions from MERRA-2 aerosol optical depth data is detailed in Sect.3.7.

Line 292-312:

3.7 MERRA-2 a priori information

Aerosol optical depth (AOD) can be expressed as the product of aerosol number concentration and the particle extinction cross section. For a given aerosol component, the column AOD at wavelength $\lambda$ can be written as

$$AOD(\lambda) = \int N(z) \cdot \sigma\_ext(\lambda) \, dz,$$

where $N(z)$ is the particle number concentration at height z, and $\sigma\_ext(\lambda)$ is the extinction cross section of an individual particle.

The column-integrated number concentration $N\_col$ is defined as the vertical integral of the particle number concentration,

$$N\_col = \int N(z) \, dz.$$

Substituting this definition into the expression above yields

$$AOD(\lambda) = N\_col \cdot \sigma\_ext(\lambda).$$

The extinction cross section can be further expressed as

$$\sigma\_ext(\lambda) = Q\_ext(\lambda) \cdot \pi \cdot r\_eff^2,$$

where Q_ext is the dimensionless extinction efficiency and r_eff is the effective particle radius.

Combining these expressions gives a first-order estimate of the aerosol column number concentration,

$$N\_col = AOD(\lambda) / [Q\_ext(\lambda) \cdot \pi \cdot r\_eff^2].$$

For each aerosol component in MERRA-2, representative values of the effective radius r_eff are adopted following the aerosol size assumptions used in the GEOS model. Accumulation-mode aerosols, such as sulfate and organic carbon, typically have effective radii on the order of $r\_eff \approx 0.3–0.4$ μm, while other aerosol components exhibit component-dependent sizes. The corresponding extinction efficiencies Q_ext(λ) are obtained from MOPSMAP simulations using consistent refractive indices and particle size assumptions.

Finally, the estimated column number concentrations are normalized across aerosol components to obtain relative number concentration ratios. These ratios are used as prior information in the retrieval.

Comment 5

Line 105: which wavelength(s) are SSA and AF used at? From table 1 I think just 440 nm, this should be specified here as well.

Response 5

Yes, thanks for your suggestion. SSA and AF are at 440 nm. We add this in the text.

Comment 6

Line 138: I think this sentence as written is too optimistic and would say that these are "sometimes" available from "ground-based" remote sensing observations. They are rarely available robustly from satellite retrievals, especially for the bulk of scenes where AOD is low. And from ground-based remote sensing like AERONET, they are also much more limited than direct-Sun data due to a less frequent measurement cadence and various other scene requirements (e.g. azimuthal symmetry which limits the applicability in cases of plumes where the aerosol is spatially hetereogeneous).

Response 6

We agree with you and have made the following revisions as you suggested:

Line 126: Specifically, the ML model takes as input the aerosol component fractions (i.e., number concentrations of sea salt, sulfate, black carbon, dust, and insoluble aerosols), together with auxiliary parameters such as single scattering albedo (SSA), asymmetry factor (AF), and effective radius (Reff). These parameters are not routinely available from all remote sensing observations: they are only sometimes retrievable from ground-based measurements under specific viewing and scene conditions, and are rarely robustly retrieved from satellite observations, particularly in low-AOD scenes or in cases where the aerosol field is spatially heterogeneous.

Comment 7

Line 191: I still don't see why using MOPSMAP directly should result in worse results than the NN emulator. To me it suggests a code bug or an issue with e.g. minimization settings (maybe something numerical which becomes less of an issue in the NN normalization process). The underlying reality is the same and if the code is written well then it should not care whether it is being fed MOPSMAP or an emulator. So I am still suspicious that the NN emulator is needed at all, though it is of course useful as a speedup.

Response 7:

Thank you for raising this important point. We added the additional experiments and analyses presented in Appendix A, which are intended precisely to clarify this issue. Our results demonstrate that the difference between using MOPSMAP directly and using the NN-based forward model does not arise from numerical instability, code implementation, or minimization settings, but from the dimensionality of the inversion state vector and the intrinsic information content of the observations.

When MOPSMAP is used directly as the forward model, the inversion naturally requires a high-dimensional microphysical state vector (15 parameters in our setup, including mode radii, size distribution widths, and number concentrations). The results in Appendix A show that, even under idealized conditions with synthetic observations generated by the same forward model, the degrees of freedom for signal (DOFS) is only about 1–2 for this full state vector. Most microphysical parameters are therefore unconstrained by the available optical observations, and the inversion is fundamentally under-determined. This behavior is reflected in the averaging kernel analysis, which shows that the majority of state vector elements are dominated by the prior rather than by the measurements.

Importantly, we also used the fixed aerosol size experiment in Appendix A to see that, if the aerosol size distribution is assumed to be known, the dimensionality of the state vector can be substantially reduced and the retrievable information on aerosol composition increases or not. This naturally raises the question of whether further information could be gained if additional optical properties, such as SSA and AF treated as inputs. In a fully physical forward model like MOPSMAP, however, these quantities are unknown outputs of the model rather than known inputs. Therefore, they cannot be fixed independently in the inversion. This limitation prevents the physical model from exploiting externally available information on SSA or AF.

By contrast, the machine-learning–based forward model adopted in the main text is constructed as a data-to-data mapping, allowing SSA, AF, effective radius, and other optical parameters to be incorporated explicitly as inputs. In this way, microphysical complexity remains embedded in the learned mapping, while the inversion state vector is aligned with the actual information content of the observations. This explains why the ML-based forward model enables a stable and informative retrieval, whereas a purely physical forward model becomes ineffective when applied to a high-dimensional microphysical state space.

Comment 8
Equation 2: I would move lines 200-202 here instead of where they currently are. I was initially confused because Table 1 gives not just AOD but also Reff, SSA, and AF as outputs but in the sense of this section, they are inputs. But then line 200 explains where the 9 inputs come from and what the outputs are.

Response 8:

Thanks, we agreed with you. We've revised here as you suggested.

Comment 9

Line 205: the citation for Adam optimization should be given. Yes it's common, but the ReLU citation is given and that is also common...

Response 9:

Thank you. We've revised here as suggested.

Kingma, D. P. and Ba, J.: Adam: A Method for Stochastic Optimization, https://arxiv.org/abs/1412.6980, 2017.

Comment 10

Equation 12 and sections 3.5 and 4.3: it looks like uncertainties in the model parameters (theta, i.e. SSA, AF, Reff, RH) are not accounted for in the cost function, or in Sy. This is essentially making the assumption that the constraint on these (from AERONET or MERRA2) is perfect, which is obviously unrealistic. For example, even in good conditions the AERONET SSA uncertainty is about 0.03. Given most of the time SSA various from about 0.8 to 1, this is not a small fraction of the range. Looking at e.g. the early Dubovik 2002 aerosol type paper (https://doi.org/10.1175/1520-0469(2002)059<0590:VOAAOP>2.0.CO;2 )one can see that an envelope of +/- 0.03 can encompass e.g. smoke, dust, urban aerosol, and marine dependent on where it falls. And that asymmetry factor shows small distinction between types at 440 nm so may have limited value. This means that the error budget is incomplete and, as a result, analyses based on the propagated uncertainties will be overconfident (the results seem better than they really are). This should at minimum be clearly acknowledged in the manuscript. Again, it is misleading about the fidelity of the results otherwise. Ideally, these uncertainties should be accounted for. One method to do so would be to perturb the four parameters in theta to simulate realistic errors (which might be different from MERRA2) and incorporate this either into Sy (via seeing how it affects calculated AOD in the MOPSMAP emulator). Incorporating model parameter error like this into Sy is common within the OEM. Or, this could be added as an additional uncertainty term within Equation 16.

Response 10

Thank you very much for pointing out that. The uncertainties in auxiliary parameters (SSA, AF, effective radius, and RH) were not explicitly accounted for in the original error budget, which could lead to overconfident retrieval results. Following this comment, we have added a dedicated subsection (Sect. 3.6, "Sensitivity analysis to auxiliary parameters") to explicitly quantify the impact of these uncertainties on the retrieved aerosol composition.

In the revised manuscript, we adopt representative uncertainty ranges for each auxiliary parameter (±0.03 for SSA, ±0.05 for AF, ±20% for Reff, and ±5% for RH), consistent with reported uncertainties from AERONET and previous studies (e.g., Dubovik et al., 2002). We then propagate these uncertainties through the trained machine-learning forward model using an ensemble-based retrieval experiment, in which all auxiliary parameters are perturbed simultaneously according to Gaussian distributions and the optimal estimation retrieval is repeated 300 times.

The resulting distributions of retrieval deviations (new Fig. 5) demonstrate that auxiliary-parameter uncertainty induces only small perturbations in the retrieved aerosol composition, typically within ±5%. More specifically, different aerosol components exhibit distinct responses to perturbations in the auxiliary parameters. Sea salt and dust tend to show slightly positive deviations, whereas sulfate exhibits a weak negative deviation. In contrast, black carbon and insoluble aerosol display no systematic offset, with their deviation distributions centered near zero. These effects do not alter the identification of the dominant aerosol component.

The following part have been added as followed:

**3.6 Sensitivity analysis to auxiliary parameters**

To assess the impact of uncertainties in auxiliary optical parameters on the aerosol composition retrieval, we conducted a two-step sensitivity analysis using the trained machine-learning (ML) forward model. Aerosol composition fractions are retrieved from multi-wavelength AOD observations, while single scattering albedo (SSA), asymmetry factor (AF), effective radius (Reff), and relative humidity (RH) are treated as auxiliary inputs. In the retrieval framework, these auxiliary parameters are assumed to be fixed but subject to uncertainty. In the following analysis, we quantify how uncertainties in these auxiliary parameters propagate into the simulated AOD spectrum and subsequently affect the retrieved aerosol composition.

We adopt representative uncertainty ranges of ±0.03 for SSA, ±0.05 for AF, ±20% for Reff, and ±5% for RH. These values are consistent with typical AERONET uncertainty estimates for SSA and with commonly reported uncertainties for aerosol optical properties (e.g., Dubovik et al., 2002).

The impact of auxiliary-parameter uncertainty is quantified using an ensemble-based retrieval experiment designed to propagate realistic parameter errors. In this experiment, all four auxiliary parameters (SSA, AF, Reff, and RH) are perturbed simultaneously according to Gaussian distributions defined by their assumed uncertainty ranges, and the optimal estimation retrieval is repeated for each perturbed realization. A total of 300 realizations are performed to characterize the resulting variability in the retrieved aerosol composition. The ensemble of retrieved aerosol compositions is then summarized to assess the sensitivity of the retrieval to auxiliary-parameter uncertainty.

**4.4 Results of sensitivity analysis to auxiliary parameters**

The distributions in Fig.5 show the deviations of the retrieved aerosol component fractions relative to the baseline retrieval obtained with fixed auxiliary parameters. For all aerosol types, the perturbation-induced deviations are narrowly distributed within ±5% and centered close to zero, indicating that uncertainty in the auxiliary parameters does not introduce systematic biases in the retrieval.

The dominant aerosol components (e.g., sulfate and dust) exhibit approximately symmetric distributions with typical spreads on the order of a few percent. In contrast, minor components such as black carbon and insoluble aerosol show strongly peaked distributions near zero, reflecting both their small absolute contributions and the limited sensitivity of the spectral observations to these components. More specifically, different aerosol components exhibit distinct responses to perturbations in the auxiliary parameters. Sea salt and dust tend to show slightly positive deviations, whereas sulfate exhibits a weak negative deviation. In contrast, black carbon and insoluble aerosol display no systematic offset, with their deviation distributions centered near zero.

It indicates that if uncertainties in the auxiliary parameters (SSA, asymmetry factor, effective radius, and relative humidity) are not explicitly accounted for in the inversion, the retrieval may underestimate sea salt and dust while overestimating sulfate. However, the magnitude of these deviations remains small, and the overall aerosol composition, particularly the dominant component, is not qualitatively affected.

[Figure]

Figure 5. Distributions of relative changes in the retrieved aerosol component fractions from the sensitivity tests described in Section Sensitivity analysis to auxiliary parameters. Four auxiliary parameters (single scattering albedo at 440 nm, asymmetry factor at 440 nm, effective radius, and relative humidity) are perturbed, and deviations from the baseline retrieval are shown for each aerosol fraction. Red vertical lines indicate zero deviation.

Comment 11

Line 248: what is the justification for these prior uncertainties? It seems arbitrary. If the numbers are changed then the uncertainties and averaging kernel will change as well. If the end use is to take e.g. MERRA2 component fractions as input, then it should be determined by analysis of MERRA2 component fraction uncertainty (which I would imagine is a function of location, among other things).

Response 11:

We thank you for this important question regarding the justification of the prior uncertainties. We agree that the choice of prior uncertainty affects the posterior uncertainty and averaging kernel and therefore requires clear justification.

In this study, two different treatments of prior uncertainty are adopted for two different purposes. For the synthetic (virtual-spectrum) experiments, a fixed prior uncertainty (0.1 in relative fraction) is prescribed. The purpose of these experiments is methodological: to diagnose the intrinsic information content of the observations and to examine the behavior of the inversion framework under controlled conditions.

For the retrievals using actual observations, the prior uncertainty is not treated as a fixed or arbitrary value. Instead, it is derived empirically from MERRA-2 data and is location dependent. Specifically, for each observation site, we compute the standard deviation of the MERRA-2 aerosol component fractions within a latitude–longitude box of ±1° centered on the observation location. This spatial variability is then used as a proxy for the prior uncertainty, reflecting regional heterogeneity in aerosol composition and addressing the reviewer's concern that prior uncertainty should depend on location.

The a priori covariance matrix Sa is specified as a diagonal matrix. For the synthetic (virtual-spectrum) experiments, a uniform variance of 0.01 (i.e., a standard deviation of 0.1) is prescribed for each aerosol component. For retrievals using actual observations, the diagonal elements of Sa are derived empirically from the spatial variability of MERRA-2 aerosol component fractions within a ±1º latitude-longitude box centered on the observation location, and are used as a proxy for location-dependent prior uncertainty.

Comment 12

Line 252: Note the AERONET team report direct-Sun AOD uncertainty of 0.02 at 440 nm and shorter wavelengths. 0.01 as used here is for the longer visible wavelengths. So this should be updated.

Response 12

Corrected as recommended.

For the measurement error covariance matrix Sy, we distinguish between visible and shortwave infrared (SWIR) wavelengths. For visible bands, we adopt wavelength-dependent AOD uncertainties following AERONET direct-Sun uncertainty estimates, with a standard deviation of 0.02 at 440 nm and shorter wavelengths, and 0.01 at longer visible wavelengths.

Comment 13.

Section 4.5: if I understand correctly, in this section Reff, SSA, AF, and RH are switched around to be retrieved and not assumed. But none of these results are shown, and this is inconsistent with the rest of the retrieval development and analysis in the paper. It is also not clear why the authors chose this as opposed to just using auxiliary inputs like they did elsewhere in the paper. Line 381 sounds honestly like an excuse like the authors did not want to download the MERRA2 data needed for the case study. I also previously had concerns about the reasonableness of using monthly data for this purpose (it ignores sampling issues and real sub-monthly variability). In my view a simple monthly plot and then showing dust AOD is not convincing enough. So I do not think this section as presented is very useful in the context of the paper. My recommendation would be to keep the same retrieval methodology (i.e. auxiliary Reff, SSA, AF, and RH) as elsewhere in the paper, and to do the analysis using daily instead of monthly data. Then the data could be compared on a daily basis with the GEOS fields (sampled around the early-pm satellite overpass time) and a commonly-sampled dust AOD (and AOD for the other aerosol types) could also be presented here. This would be much more direct and meaningful demonstration of the method.

Response 13

As you correctly point out, the inversion strategy within the satellite differs from the primary methodology, treating Reff, SSA, AF, and RH as inversion variables rather than auxiliary inputs. Moreover, as you suggest, meaningful satellite validation necessitates ensuring consistency in spatio-temporal sampling.

We concur with your recommendation that inversion should employ identical methodologies during validation to guarantee consistency and clarity of results. We further agree that existing monthly averaging overlooks submonthly variability and sampling issues, and as the reviewer suggests, meaningful satellite validation must ensure consistency.

Therefore, without compromising the overall methodology or conclusions in our manuscript, we have decided to remove this satellite validation part from the main context. The revised version will retain the original auxiliary parameter inputs (Reff, SSA, AF, and RH) and focus entirely on ground applications, thereby enabling more precise constraints and validation.

We appreciate your suggestions and will undertake more precise and detailed satellite validation in future work. Building upon the current methodology, we will update and validate its feasibility and uncertainty in satellite inversion applications.

Response 14

We have removed all results concerning satellites, so this part has been deleted accordingly.

Response 14

We have removed all results concerning satellites, so this part has been deleted accordingly.

---

## Author Response (AR3)

Comment: I was a reviewer of the previous versions of this manuscript. The authors have addressed my remaining concerns sufficiently well in this revision. I think the decision to remove the satellite section was sensible. I have only one new issue: the manuscript cites Giles et al (2019a) and Giles et al (2019b), these are the same paper in the bibliography. I assume this is an error in the authors' reference management software. Once corrected I feel this manuscript is publishable in AMT.

Response to Reviewer:
We sincerely thank the reviewer for the evaluation of our work. The duplicate reference to Giles et al. (2019a,b) has been corrected, and only a single entry now appears in the reference list. Thank you for noting this issue.